# A model for interaction between conduits and surrounding hydraulically connected distributed drainage based on geomorphological evidence from Keewatin, Canada.

Emma L.M. Lewington[1], Stephen J. Livingstone[1], Chris D. Clark[1], Andrew J. Sole[1] and Robert D. Storrar[2]

[1] Department of Geography, University of Sheffield, Sheffield, UK

[2] Department of Natural and Built Environment, Sheffield Hallam University, Sheffield, UK

## Abstract

We identify and map visible traces of subglacial meltwater drainage around the former Keewatin Ice Divide, Canada from high resolution Arctic Digital Elevation Model (ArcticDEM) data. We find similarities in the characteristics and spatial locations of landforms traditionally treated separately (i.e. meltwater channels, meltwater tracks and eskers) and propose that creating an integrated map of 'meltwater routes' captures a more holistic picture of the large-scale drainage in this area. We propose the grouping of meltwater channels and meltwater tracks under the term 'meltwater corridor' and suggest that these features in the order of 10s – 100s m wide, commonly surrounding eskers and transitioning along flow between different types, represent the interaction between a central conduit (the esker) and surrounding hydraulically connected distributed drainage system (the meltwater corridor). Our proposed model is based on contemporary observations and modelling which suggest that connections between conduits and the surrounding distributed drainage system within the ablation zone occur as a result of over pressurisation of the conduit. The widespread aerial coverage of meltwater corridors (5 – 36 % of the bed) provides constraints on the extent of basal uncoupling induced by basal water pressure fluctuations. Geomorphic work resulting from repeated connection to the surrounding hydraulically connected distributed drainage system suggests that basal sediment can be widely accessed and evacuated by meltwater.

## 1. Introduction

Variations in the configuration of subglacial hydrological systems are key to understanding some of the most dynamic ice sheet behaviour at a range of spatial and temporal scales (e.g. Zwally et al., 2002; Das et al., 2008; Joughin et al., 2008; van de Wal et al., 2008; Shepherd et al., 2009; Palmer et al., 2011; Fitzpatrick et al., 2013; Doyle et al., 2014). Once water reaches the bed, its impact on ice flow is determined by the hydraulic efficiency of the subglacial hydrological system. Theory developed at alpine glaciers suggests that increasing water pressure results in enhanced ice motion owing to reduced ice-bed contact (Lliboutry, 1968; Bindschadler, 1983) and where sediment is present, enhanced sediment deformation (e.g. Engelhardt et al., 1978; Hodge, 1979; Iken and Bindshadler, 1986; Fowler, 1987; Iverson et al., 1999; Bingham et al., 2008). Water pressure at the bed depends on water supply to, storage within and discharge through the subglacial hydrological system (Iken et al., 1983; Kamb et al., 1985; Nienow et al., 1998). The configuration of the subglacial hydrological system is key to this, with a hydraulically efficient drainage system able to accommodate and evacuate an equivalent water flux without causing spikes in basal water pressure which have been linked to transient ice accelerations (e.g. Tedstone et al., 2013).

Traditionally the subglacial hydrological system has been conceptualised as a binary model comprising (i) inefficient distributed drainage - taking the form of thin films of water (Weertman, 1972), linked cavities (Lliboutry, 1986; Walder, 1986; Kamb, 1987), groundwater flow (Boulton et al., 1995) and / or wide shallow canals (Walder and Fowler, 1994); and (ii) efficient channelised drainage with conduits cut either up into the ice (Rothlisberger-channel) or down into the bed (Nye-channel) (e.g. Rothlisberger, 1972; Shreve, 1972; Nye,1973; Hooke et al., 1990). These two systems interact with each other over a range of spatial and temporal scales (e.g. Andrews et al., 2014; Hoffman et al., 2016; Rada and Schoof, 2018; Downs et al., 2018; Davison et al., 2019), resulting in: (i) a moulin-connected channelised system which remains hydraulically connected to surface meltwater inputs throughout the melt season; (ii) an active hydraulically connected distributed system strongly influenced by the channelised system and therefore surface inputs across a range of spatial and temporal scales (e.g. Hubbard et al., 1995) and; (iii) a weakly-connected distributed system largely isolated from the channelised system and only rarely – if ever - affected

by surface meltwater inputs (Andrews et al., 2014; Hoffman et al., 2016; Rada and Schoof, 2018).

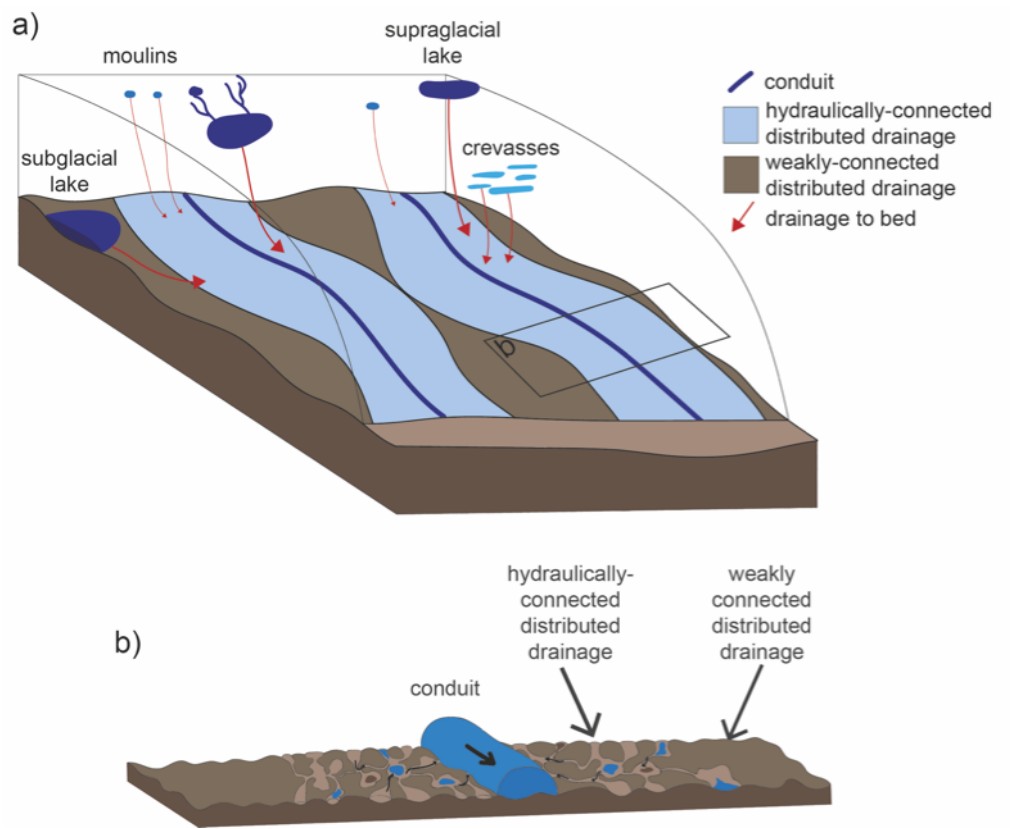

**Figure 1.** Ice sheet hydrological system with varying surface and basal inputs (a) and a three-
system drainage model (b). In the three-system drainage model, the hydraulically connected distributed drainage system (light blue in a), is influenced regularly by surface meltwater inputs through the conduit. The weakly-connected distributed drainage system (dark brown) is largely isolated and rarely or never impacted by surface meltwater inputs. At present, the relative coverage of each is not yet known, nor the precise configuration or relation between each
component.

     In theory, in a steady-state system, water flows from surrounding high pressure distributed regions into lower pressure conduits. Borehole measurements of subglacial
water pressure, modelling and ice velocity proxy data (e.g. Hubbard et al., 1995; Gordon et al., 1998; Bartholomaus et al., 2008; Werder et al., 2013; Tedstone et al., 2014) suggest, however, that given a sufficiently large and rapid spike in water delivery to a subglacial conduit, the hydraulic gradient can be reversed such that water is

forced out of and laterally away from the conduit into the hydraulically connected distributed drainage system. This has been variously termed a Variable Pressure Axis (VPA) (Hubbard et al., 1995), efficient subsystem (Rada and Schoof, 2018), efficient core (e.g. Davison et al., 2019) etc. Here, we use the term hydraulically connected distributed drainage which we consider to be the lateral limit of the influence of pressure variations that originate in a subglacial conduit and cause the flow of water in or out of the conduit. This mechanism has implications for overlying ice sheet dynamics, for example, overpressurisation overwhelms the conduit and can elicit ice flow acceleration and ice sheet surface uplift (e.g. van de Wal et al., 2008; Bartholomew et al., 2011; Doyle et al., 2013; Tedstone et al., 2015). The extent of this dynamic effect is much greater than the area of the bed directly affected by the meltwater.

Beyond the hydraulically connected distributed drainage system, the remaining distributed drainage system – likely composed of linked cavities – is largely isolated or disconnected from surface meltwater inputs (Andrews et al., 2014; Hoffman et al., 2016; Rada and Schoof, 2018). This area may exhibit some slow leakage into the hydraulically connected distributed drainage system (Hoffman et al., 2016; Rada and Schoof, 2018) and it is possible that pressure perturbations within the conduit also increase connections between these weakly connected parts of the bed and the hydraulically connected distributed drainage system. Weakly connected drainage areas potentially cover a large percentage of the bed and their gradual drainage over time is hypothesised to reduce regional basal water pressure, thereby increasing ice-bed contact and reducing ice velocity (e.g. Sole et al., 2013; Andrews et al., 2014; Bougamont et al., 2014; Tedstone et al., 2015; Hoffman et al., 2016).

Although we now have a better appreciation for the heterogeneous nature of the subglacial hydrological system, a lack of direct observations means that the reality of this interaction – its spatial and temporal occurrence, its expression and impact – remains speculative. For example, is the transition between the connected and isolated parts of the distributed drainage system abrupt (e.g. Hoffman et al., 2016) or transitional (e.g. Hubbard et al., 1995; Downs et al., 2018); how does the extent of the hydraulically connected distributed drainage system vary over space and time; does

the forcing of pressurised water out of the conduit have implications for sediment erosion rates?

## 1.1 Palaeo-meltwater landforms

Palaeo–meltwater landforms have been fundamental in inspiring and guiding conceptual and numerical models of how water self-organises into drainage systems beneath present day ice masses because they can be easily observed and investigated (Fig. 2). Such landforms are therefore key to contextualising spatially and temporally limited contemporary observations and are commonly used to support and develop the theory of ice sheet hydrological systems (e.g. Shreve, 1985; Clark and Walder, 1994; Boulton et al., 2007a, 2007b; 2009; Beaud et al., 2018a; 2018b; Hewitt and Creyts, 2019). Much of this focus has been on landforms such as eskers, meltwater channels and tunnel valleys which indicate efficient channelised subglacial drainage (e.g. Shreve, 1985; Brennand, 1994, 2000; Clark and Walder, 1994; Punkari, 1997; Boulton et al., 2007a, 2007b, 2009; Storrar et al., 2014a; Livingstone and Clark, 2016). We will now discuss each of these in more detail.

### 1.1.1 Eskers

Eskers are linear depositional landforms made up of glaciofluvial sand and gravel deposited from meltwater flowing through or beneath an ice mass in conduits metres to tens of metres in width and height. They exist as individual segments that often align to form networks extending up to several hundreds of kilometres (e.g. Shreve, 1985; Aylsworth and Shilts, 1989; Brennand, 2000; Storrar et al., 2014a; Stroeven et al., 2016) and are typically taken to record the former position and characteristics of Röthlisberger-channels (R-channels) thermally eroded into the base of the ice by turbulent water flow. While most studies reduce esker mapping to a single crest-line and consider the 'classic' single straight-to-sinuous undulating ridge to be pervasive, more complex esker morphologies also occur (e.g. Banerjee and McDonald, 1975; Rust and Romanelli, 1975; Hebrand and Amark, 1989; Gorrell and Shaw, 1991; Warren and Ashley, 1994; Brennand, 2000; Mäkinen, 2003; Perkins et al., 2016; Storrar et al., 2019). These include fine-grained sandy fan shape elements or 'splays', alongside and associated with the coarse gravelly central ridge (e.g. Cummings et al.,

2011a; Prowse, 2017). These splays are an order of magnitude wider and more gently sloped than the main ridge (Cummings et al., 2011a). They are proposed to form in proglacial environments, representing subaqueous outwash fans deposited by sediment laden plumes exiting a subglacial conduit into a proglacial lake (e.g. Powell, 1990; Hoyal et al., 2003; Cummings et al., 2011b), supraglacial environments (e.g. Prowse, 2017) and subglacial environments, with sedimentation in subglacial cavities alongside the main esker ridge during periods of high water pressure within the conduit (e.g. Gorrell and Shaw, 1991; Brennand, 1994).

### 1.1.2  Meltwater channels and tunnel valleys

Erosional subglacial meltwater channels, or Nye-channels (N-channels), incised into bedrock or sediment substrate range in size from metres to tens of metres wide (e.g. Sissons, 1961; Glasser and Sambrook Smith, 1999; Piotrowski, 1999) to large tunnel valleys several kilometres in width and tens of kilometres long (e.g. Kehew et al., 2012; van der Vegt et al., 2012; Livingstone and Clark, 2016). Tunnel valleys are observed to occur at various developmental stages from mature and clearly defined to indistinct valleys often associated with hummocky terrain or as a series of aligned depressions (e.g. Kehew et al., 1999; Sjogren et al., 2002). Their formation has been linked to subglacial meltwater erosion at the ice-bed interface (c.f. Ó Cofaigh, 1996; Kehew et al., 2012; van der Vegt et al., 2012) with the assumption that channels transported large volumes of sediment and water. However, their precise mechanism of formation is still debated with the main arguments focussing on i) catastrophic outburst formation with rapid erosion following the release of sub or supraglacially stored water (e.g. Piotrowski, 1994; Cutler et al., 2002; Hooke and Jennings, 2006; Jørgensen and Sandersen, 2006); ii) gradual steady-state formation with headward erosion of soft-sediments in low water pressure conduits (e.g. Boulton and Hindmarsh, 1987; Mooers, 1989; Praeg, 2003; Boulton et al., 2009); and iii) formation from seasonal meltwater flow (Beaud et al., 2016, 2018b).

Here, we use the term meltwater channel to refer to palaeo-evidence of erosional channelised flow preserved on the ice sheet bed (i.e. the outline of the path the water took) at all scales from N-channels through to tunnel valleys. We use the term conduit

to refer to the active channelised flow beneath a contemporary ice mass (i.e. the enclosed (sediment or ice walled) pipe carrying water at the ice-bed interface).

### 1.1.3 Meltwater tracks

Detailed mapping in northern Canada and Scandinavia has identified the presence of linear tracks variously termed 'hummock corridors', 'glaciofluvial corridors', 'washed zones' and 'esker corridors', typically a few hundred meters to several kilometres wide and a few kilometres to hundreds of kilometres long (e.g. St-Onge, 1984; Dredge et al., 1985; Rampton, 2000; Utting et al., 2009; Burke et al., 2012; Kerr et al., 2014a, 2014b; Sharpe et al., 2017; Peterson et al., 2017; Peterson and Johnson, 2018; Lewington et al., 2019). These features often contain eskers and hummocks which vary in size, shape and relief (Peterson and Johnson, 2018) as well as 'patches' of glaciofluvial deposits and areas of exposed bedrock. While a subglacial meltwater origin is largely agreed upon, their precise mode of formation is not yet known. These features are collectively termed meltwater tracks hereon in.

Meltwater landforms are typically mapped and interpreted individually (e.g. Clark and Walder, 1994; Brennand, 2000; Storrar et al. 2013; Burke et al., 2015; Livingstone and Clark, 2016; Mäkinen et al., 2017) rather than as a holistic drainage signature (c.f. Storrar and Livingstone, 2017). As such, it is not yet clear whether or how differing expressions of subglacial drainage are interrelated and to what extent variations in drainage or background conditions (e.g. bed substrate, geology and local topography) control the preserved geomorphic signature we see today. This study aims to identify and map all discernible evidence of subglacial meltwater drainage across the Keewatin District of northern Canada from the ArcticDEM. We collectively refer to these as meltwater routes. Producing an integrated map of all visible subglacial meltwater evidence allows us to quantify the varying dimensions and geomorphological expressions of these features, to investigate associations between features traditionally treated separately and to explore potential controls on expression and formation. Importantly, we note this is a minimum map as some landforms – particularly tunnel valleys – may be fully or partially buried (e.g. Jørgensen and Sandersen, 2006).

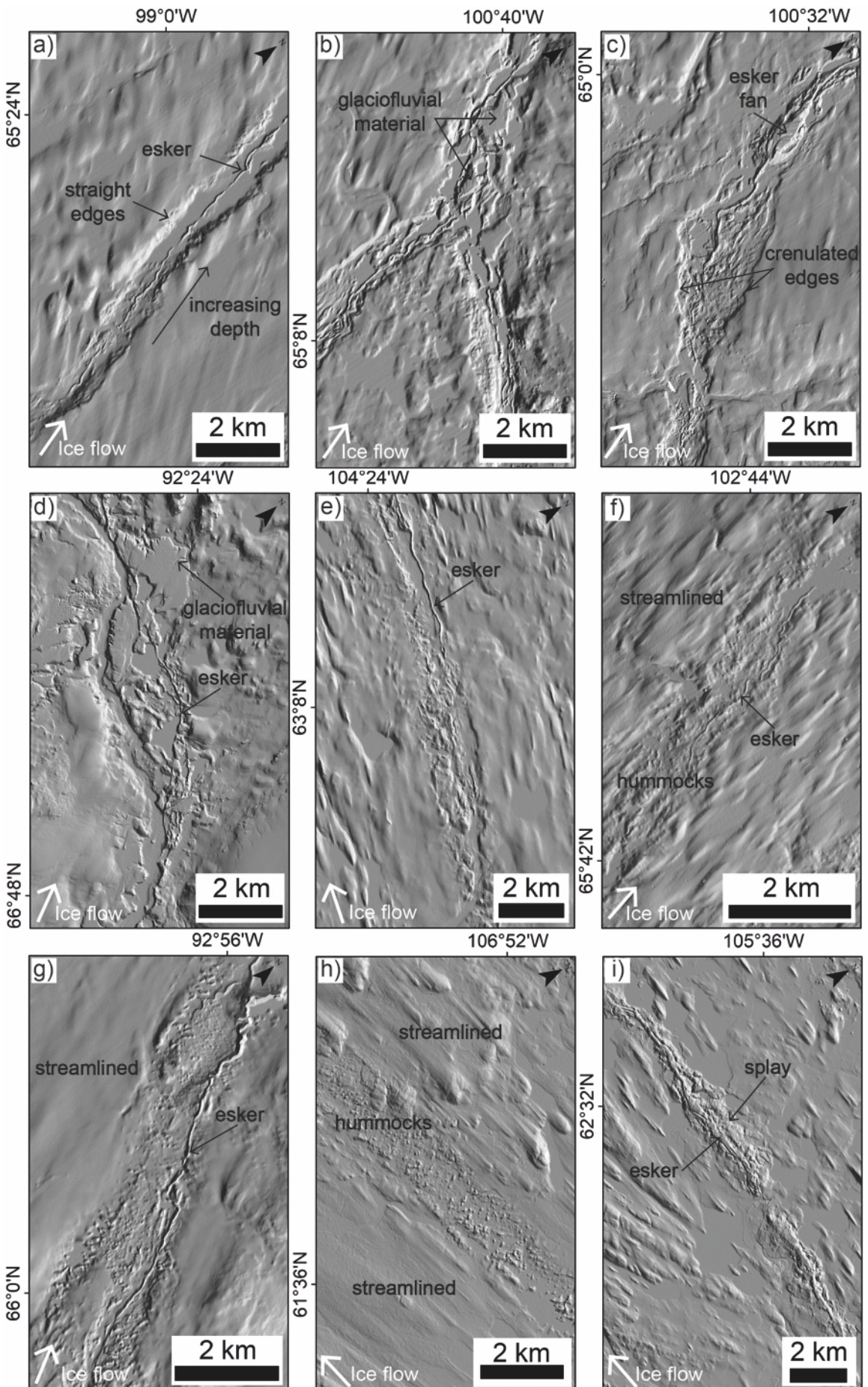

**Figure 2** (above). Varying geomorphic expressions of subglacial meltwater flow: (a) increasing depth hummock corridor transitioning into a tunnel valley; (b-d) hummock corridors with negative relief containing eskers, esker fans and glaciofluvial deposits. Corridor edges vary in straightness; (e-h) hummock corridors with more subdued relief, largely detectable by the elongated tracts of hummocks which stand out from the surrounding streamlined terrain and often surround an esker; (i) an esker surrounded by lateral fans i.e. esker splays.

## 2. Study Area

This study focusses on an area approximately 1 million km$^2$ to the west of Hudson Bay in northern Canada, surrounding the location of the former Keewatin Ice Divide of the Laurentide Ice Sheet (LIS) (Fig. 3) (Lee et al., 1957; McMartin et al., 2004). The area generally exhibits negligible local relief and is underlain by resistant Precambrian bedrock that is either exposed or covered by till ranging from thin and discontinuous (typically < 2 m) to thick and pervasive (typically > 2 m) (e.g. Clark and Walder, 1994).

Traditionally, eskers have been identified as the predominant meltwater landform within the Keewatin area, although meltwater tracks (e.g. St-Onge, 1984; Aylsworth and Shilts, 1989; Rampton, 2000; Utting et al., 2009; Sharpe et al., 2017; Lewington et al., 2019) and meltwater channels (e.g. Storrar and Livingstone, 2017) have also been recorded. At a large scale, eskers radiate out from the ice divide, beneath which they are rare (Shilts et al., 1987; Aylsworth and Shilts, 1989; Storrar et al., 2013, 2014a). At a local to regional scale, they exhibit a dendritic pattern and 12–15 km quasi-uniform spacing (e.g. Banerjee and McDonald, 1975; St-Onge, 1984; Shilts et al., 1987; Bolduc, 1992; Storrar et al., 2014a).

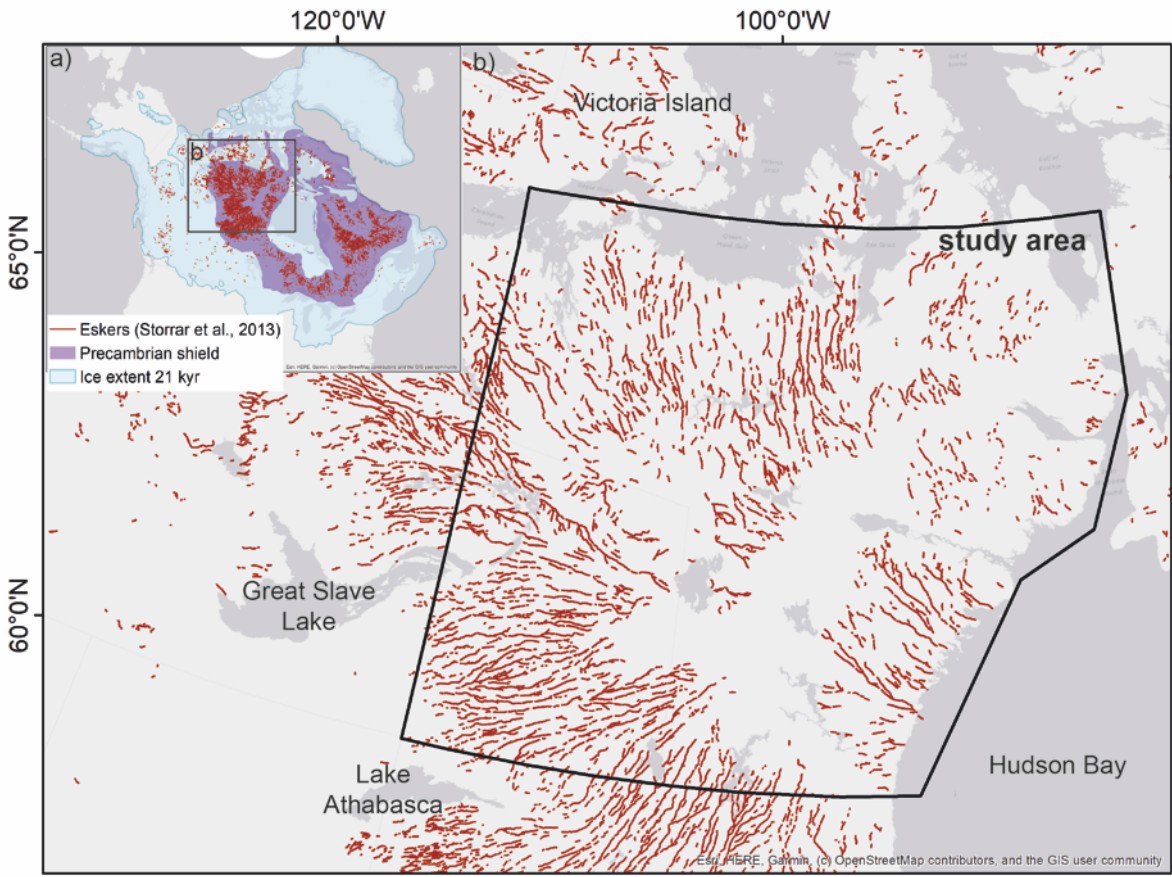

**Figure 3.** Large-scale distribution of eskers around Hudson Bay (Storrar et al., 2013) (a). The Laurentide Ice Sheet extent displayed in the inset is the Last Glacial Maximum (LGM) at 18 $^{14}$C ka BP (21.4 cal. ka BP) (Dyke et al., 2003) and the extent of the Precambrian shield is also mapped (Wheeler et al., 1996). Zoomed in location of the study area focussed on the area around the former Keewatin Ice Divide (b).

## 3. Methods

### 3.1 Data sources and mapping

High-resolution digital elevation data, made available through the ArcticDEM (10 m) (freely available at https://www.pgc.umn.edu/data/arctcidem), and generated by applying stereo and auto-correlation techniques to overlapping pairs of high-resolution optical satellite images (Noh and Howat, 2015; Porter et al., 2018), were used in this study to identify and map meltwater landforms. In addition, eskers mapped by Storrar et al., (2013) from 30 m resolution Landsat ETM+ multispectral imagery

were used to inform further high-resolution esker mapping from the 10 m resolution ArcticDEM. The automatic mapping approach developed in Lewington et al. (2019) was used to create a first pass map of hummock corridors – classified as meltwater tracks here (Appendices Fig. A1) - to augment the improved esker map. Together, these were used to create an integrated map of meltwater routes by manually mapping centrelines of all visible traces of subglacial meltwater drainage including meltwater tracks, meltwater channels and eskers. Multiple orthogonal hillshades were generated to avoid azimuth bias (Smith and Clark, 2005) and mapping was undertaken at a range of spatial scales to maximise the number of features captured (Chandler et al., 2018).

## 3.2. Classification and morphometry

The meltwater routes were used to explore the occurrence and morphology of different types of meltwater landforms. Former ice-margin estimates from Dyke et al. (2003) were used as transects (Fig. 4). These transects are spaced approximately 30 – 40 km apart and in the study area, cover c. 1,000 years of deglaciation between 9.7 and 8.6 ka. This period encompasses the final stages of deglaciation when the ice sheet was experiencing a strongly negative surface mass balance with associated increasing rates of meltwater production (e.g. Carlson et al., 2008, 2009). Retreat rates were generally between 100 - 200 m yr$^{-1}$ from 13 to 9.5 ka, increasing rapidly between 9.5 and 9 ka to around 400 m yr$^{-1}$ after which retreat rate decreased briefly before another increase from ~8.5 ka (Dyke et al., 2003).

When a meltwater route intersected a transect, an intersection point was added, and the following information recorded:

- Landform type (i.e. esker ridge, esker with lateral splay, meltwater track or meltwater channel)
- Width of landform (or landforms if an esker ridge was present within a meltwater track, meltwater channel or surrounded by a lateral splay)
- Bed substrate and geology (Fulton, 1995; Wheeler et al., 1996).

Spacing between adjacent meltwater route centrelines was calculated along each transect with centrelines at the end of each transect and those separated by clear

breaks (e.g. due to the coincidence of a lake) discounted. The total length of meltwater route centrelines was calculated automatically in a GIS.

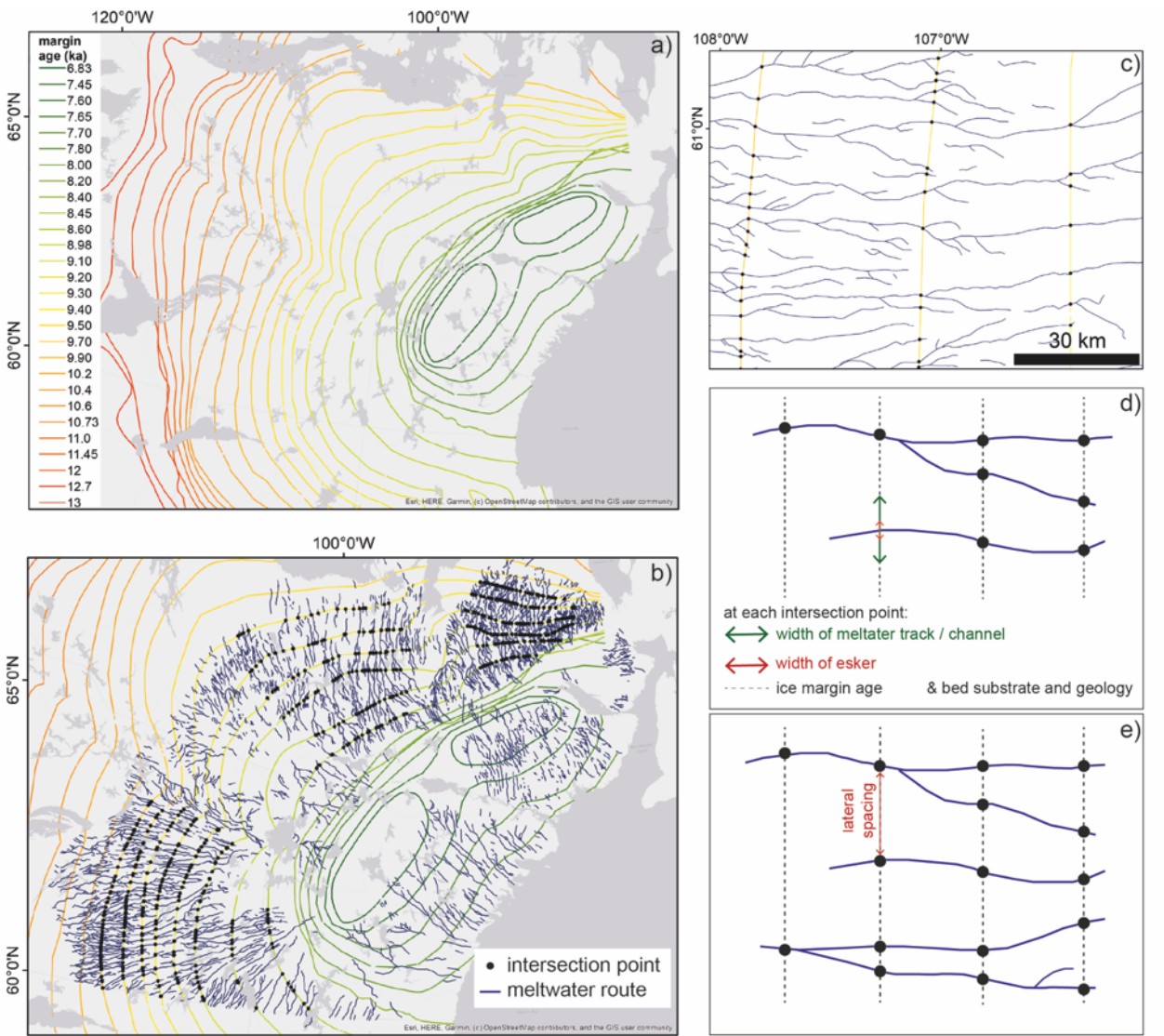

Figure 4. (a) Ice margin estimates (Dyke et al., 2003) for the Keewatin sector of the LIS; (b) Intersection points between mapped meltwater routes and ice margin estimates used for sample locations; (c) A zoomed in example of meltwater routes, margin isochrones and intersections from the SW of the study area; (d) Method for recording meltwater route characteristics and; (e) lateral spacing.

### 3.3 Testing controls on meltwater routes width and expression

This study takes a large-scale approach to exploring controls on meltwater route width and expression. While this approach results in a compromise in terms of data resolution available for the surface substrate and geology maps, it also increases statistical confidence in the results due to the larger sample size. Before analysis was undertaken, three test sites were selected from the study area to allow for more detailed mapping and comparisons (Appendices Figs. A2 and A3).

To explore substrate and geological controls on meltwater route occurrence, distribution and properties, the overall length of meltwater routes overlying each substrate type (Fulton, 1995) and geology (Wheeler et al., 1996) within the three test sites was calculated. The total area of each basal unit within the test sites was also calculated and values converted to percentages. Following this, the percentage area was subtracted from the percentage of meltwater routes for each individual substrate and geology type, giving a positive (over-represented) or negative (under-represented) value. Next, meltwater routes were split and classified by feature type (i.e. esker, esker with lateral splay, meltwater channel and meltwater track). The above analysis was then repeated by feature type to explore whether geomorphological expression is controlled by surface substrate or geology. It is important to note that categorisations along meltwater routes were not always independent as the same section was sometimes coded as a meltwater track and an esker with splay as often 'positive' features are situated within wider erosional corridors.

It was noted that landform type varies both across adjacent meltwater routes and along individual meltwater route centrelines. To assess any potential relationship between landform type and background controls in more detail, individual centrelines were selected and sampled with a higher frequency (1 km intervals). At each sample location the width of the meltwater track or meltwater channel, the presence or absence of an esker (and its width if present), surface substrate, bed geology and elevation were recorded.

The transfer of surface meltwater to the bed via moulins is thought to be strongly controlled and largely fixed by bed topography; ice flow over bedrock ridges

can cause elevated tensile stresses resulting in crevassing (Catania and Neumann, 2011), while the transfer of bed topography to the ice-surface preconditions where surface lakes form (e.g. Gudmundsson, 2003; Karlstrom and Yang, 2016; Crozier et al., 2018; Ignéczi et al., 2018). To investigate the spatial coincidence between subglacial meltwater pathway density and basal roughness, we initially applied a circular median filter with a 2 km diameter to the bed topography (the 10 m resolution resampled to 100 m). This was based on the understanding that bed perturbations below 1 - 3 times the ice thickness are not transferred to the surface (Gudmundsson, 2003; Ignéczi et al., 2018) and that the LIS ice thickness was typically 500 – 2000 m thick. Standard deviation was then calculated over a 20 km diameter window as per Ignéczi et al., (2018), who found this smoothing distance matched the requirements that the smoothing window should not exceed 10 times the ice thickness (Gudmundsson, 2003) while still capturing longer scale variations and dampening rapid changes in local topography (Ng et al., 2018).

Finally, ice stream locations (Margold et al., 2015) were quantitatively compared to the distribution of meltwater routes. This allowed us to determine whether or not there was a difference in expression of subglacial meltwater pathways between ice stream and non-ice stream areas.

## 4. Results

### 4.1 An integrated drainage signature

Mapping all traces of meltwater drainage reveals the ubiquity of former subglacial drainage across the study area (Fig 5). A total of ~ 3000 meltwater routes were mapped over a ~1 million $km^2$ area with a total length of almost 55,000 km. The meltwater routes exhibit a similar overall pattern to earlier esker maps (e.g. Aylsworth and Shilts, 1989; Storrar et al., 2013) radiating out from the former Keewatin Ice Divide. Greater than 90 % of mapped esker ridges in this region are estimated to occur along a meltwater route and therefore form part of the same network. In terms of the large-scale pattern, there are no obvious trends in meltwater route density, width or feature type associated with margin retreat. However, the study area only covers

approximately 1,000 years, associated with a period of intense meltwater production and rapid retreat

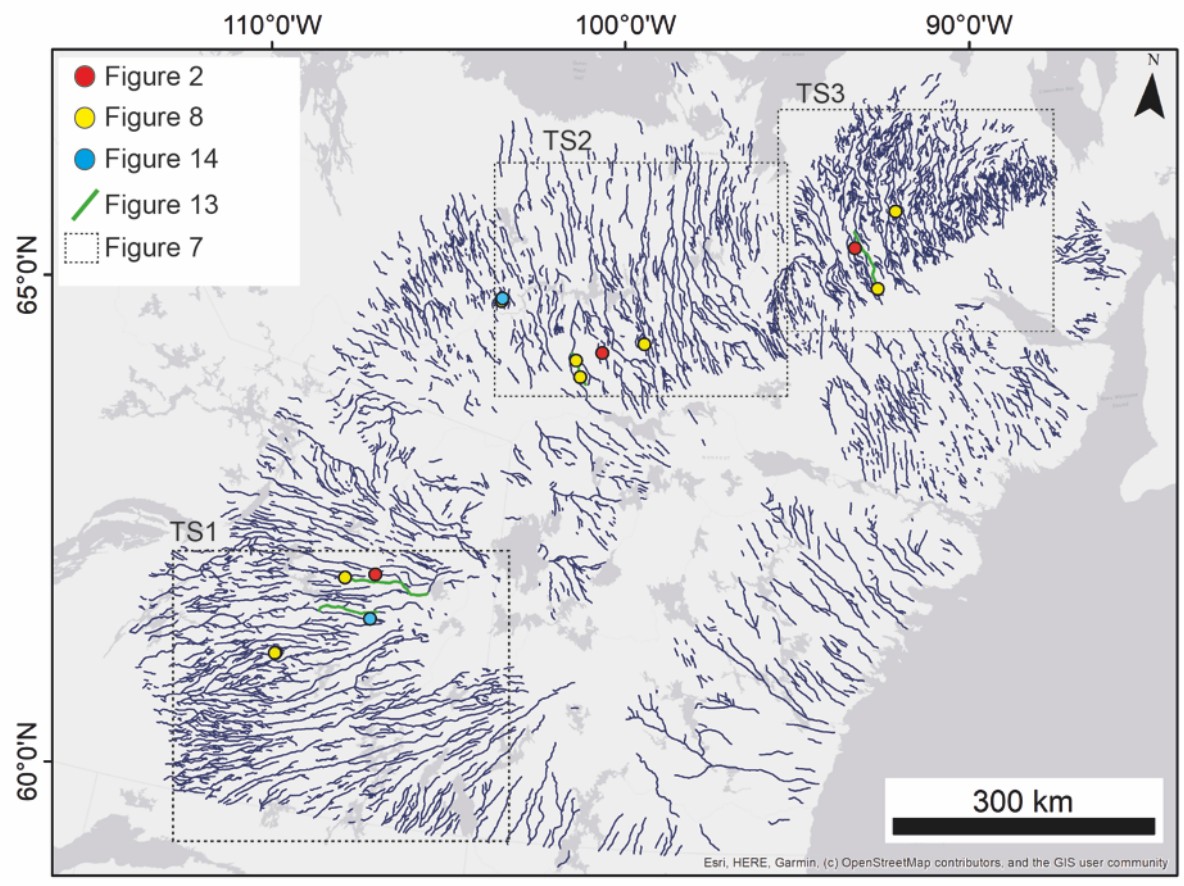

**Figure 5.** Integrated map of meltwater routes. Note how meltwater routes in this new map are less fragmented and denser than the existing esker map (Fig. 2b). Points and boxes represent locations of other figures.

Within the study area, 84 % of sample locations captured a meltwater track (65 %) or meltwater channel (19 %). The remaining samples captured an esker ridge alone (6 %), an esker ridge with a lateral splay (6 %) or were deemed unclassified (4 %). However, subglacial meltwater signatures were not always mutually exclusive and often esker ridges or sometimes even eskers with lateral splay, were recorded within the meltwater tracks and channels. Esker mapping by Storrar et al., (2013) was updated in the study area. Due to the higher resolution data available and the smaller spatial area covered, smaller features which may have been missed could be included. A comparison between the updated esker map and the new meltwater routes map confirms the large-scale association between eskers and wider meltwater features

which often flank and connect intervening segments. Eskers were recorded at 43 %
of all sample locations. Where they were recorded, 87 % of the time they were flanked
by a meltwater track, channel or splay.

**Table 1.** Summary statistics for meltwater routes in the study area.

|  | Length (km) | Width (km) | Centreline spacing (km) |
|---|---|---|---|
| Min | 0.7 | 0.05 | 0.4 |
| Lower quartile | 4.8 | 0.5 | 3.3 |
| Upper quartile | 20.1 | 1.1 | 10.1 |
| Max | 339.9 | 3.3 | 77.9 |
| Mean | 18.1 | 0.9 | 8.1 |
| Std dev. | 26.5 | 0.6 | 7.4 |

Meltwater routes reach a maximum of 3.3 km in width and 340 km in length
(Table 1), but are noted to reach up to 760 km when they extend beyond the limits of
the study area (Storrar et al., 2014a). Meltwater channels and meltwater tracks are
typically an order of magnitude wider (mean width: 900 m) than the eskers which they
often contain (mean width: 97 m). Meltwater routes appear to vary in width across the
study area and along individual centrelines but show no clear trend from the ice divide
towards the margin. If these landforms are assumed to have formed time-
transgressively, this would suggest no clear trend in width during deglaciation. Within
the study area, adjacent centrelines are spaced on average 8 km apart (Table 1). This
is at the lower end of the range reported in the literature (Fig. 6) (e.g. Banerjee and
McDonald, 1975; St-Onge, 1984; Shilts et al., 1987; Hebrand and Amark, 1989;
Bolduc, 1992; Boulton et al., 2009; Hewitt, 2011). This is not surprising given that we
mapped all traces of subglacial meltwater flow including meltwater tracks not
containing eskers. Like variations in width, there appears to be no coherent change in
spacing during deglaciation (Fig. 4) if we assume time transgressive formation.

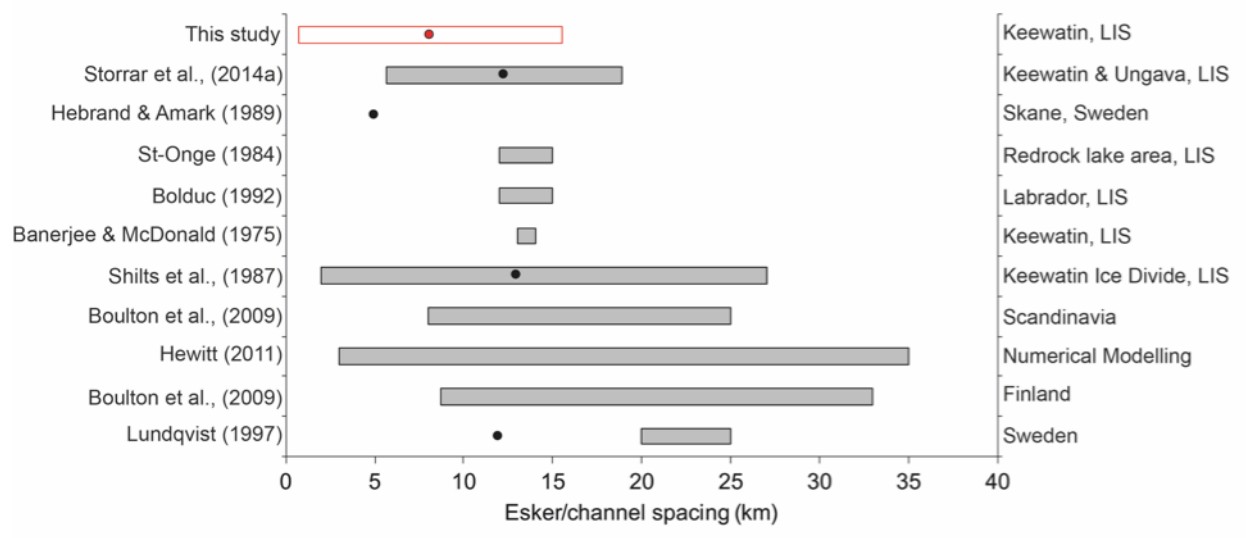

**Figure 6.** Examples of esker and subglacial channel spacing quoted in the literature with bars representing maximum and minimum and the points the mean (Storrar et al. 2014a). The top two bars represent a large-scale esker sample taken from an area which includes this study (Storrar et al., 2014a) and the spacing recorded by all visible traces of subglacial meltwater (i.e. eskers and meltwater corridors). For these two, the bars represent standard deviation and the points the mean. (Modified from Storrar et al., 2014a).

Eskers have been widely mapped in northern Canada. Initial mapping was largely undertaken by the Geological Survey of Canada using aerial photography and field observations (e.g. Aylsworth and Shilts, 1989). This included mapping of 'esker systems' – comprising a series of hummocks or short, flat-topped segments which phase downstream into relatively continuous esker ridges or occasionally beaded eskers – across 1.3 million $km^2$ of the Keewatin sector of the LIS (Aylsworth and Shilts, 1989; Aylsworth et al., 2012). Discontinuous esker ridges are connected to areas of outwash, meltwater channels or belts of bedrock stripped free of drift. More recently, increasing availability of remotely sensed data allowed Storrar et al., (2013) to digitise eskers at an ice sheet scale for the LIS (including the Keewatin sector) using Landsat 7 ETM+ imagery. From this, a secondary dataset was derived by interpolating a straight line between successive aligned esker ridges, creating a continuous pathway, which reflects the location of the major conduits in which the eskers formed (Storrar et al., 2014a). This paper extends earlier work, which recognises links between eskers and broader traces of subglacial meltwater flow but does not explicitly describe or

formally quantify them (e.g. Aylsworth and Shilts, 1989; Storrar et al., 2014a). It is encouraging that despite different datasets and mapping procedures, the overall patterns are similar (Fig. 7).

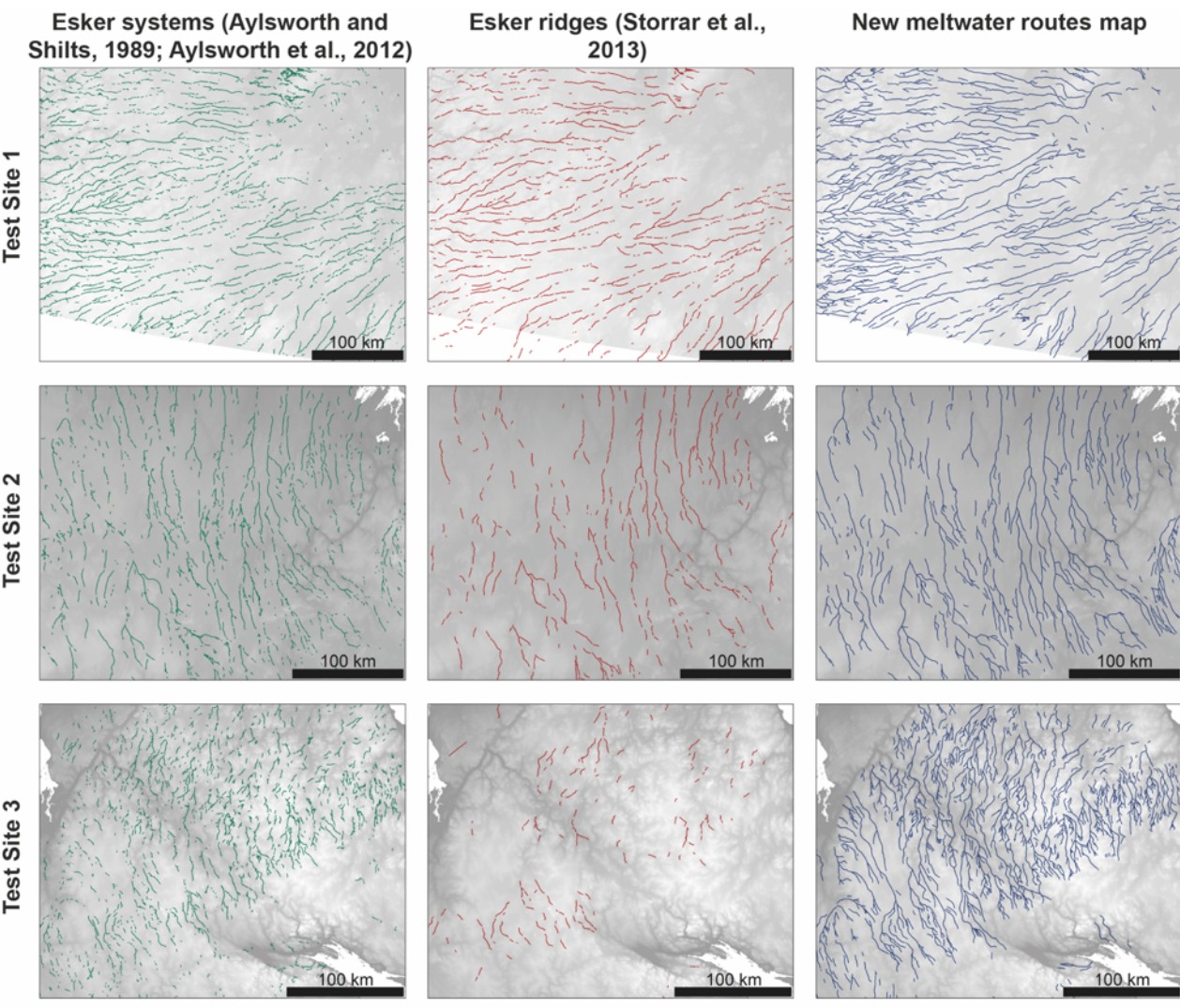

**Figure 7.** Comparison of existing maps of 'esker systems' (green) from air photo interpretation (Aylsworth and Shilts, 1989; Aylsworth et al., 2012), esker ridges (red) from Landsat imagery (Storrar et al., 2013) and the new meltwater routes from the ArcticDEM (blue). Mapping of meltwater routes includes all traces of subglacial meltwater flow (eskers, eskers with lateral splays, meltwater tracks and meltwater channels). The locations of test site 1, test site 2 and test site 3 are identified in Fig. 5. DEM(s) created from the Canadian Digital Elevation Model (CDEM). Ottawa, ON: Natural Resources Canada. [2015]

**4.3 Geomorphological variations**

Landforms along meltwater routes exhibit a high degree of geomorphic variability and each of the palaeo-meltwater landforms outlined in section 1.1 (meltwater channels, meltwater tracks and eskers) are identified in the study area. Meltwater channels exhibit negative relief down to ~ 30 m below their immediate surroundings (e.g. Fig. 2a). Meltwater tracks exhibit less pronounced (e.g. Fig. 2b - 2d) or even negligible relief (e.g. Fig, 2e - 2h), the latter being identified due to the

presence of elongated tracts of hummocks. Meltwater route edges vary from straight (e.g. Fig 2a, 2e, 2h) to crenulated (e.g. Fig. 2c) and may be discontinuous along sections. A variety of landforms are found within the meltwater tracks and channels. These include hummocks of varying size, shape and relief  (e.g. Fig 2e – 2h) and eskers and associated glaciofluvial material (e.g. Fig. 2a – 2d). In places, till may be

entirely eroded, revealing patches of bedrock. Eskers display a high degree of variability along the meltwater routes with single, continuous ridges the exception rather than the norm. Meltwater routes vary in geomorphological expression both across flow, between adjacent routes, and along flow, with multiple transitions to and from 'different' feature types (Fig. 8).


**Figure 8 (below).** Examples of transitions and associations along meltwater routes. The left panel shows the DEM and the right panel shows an interpretation of the feature types with an inset (top right) showing how meltwater routes are mapped as single lines through all types. White patches in the DEM represent areas of missing data due to the presence of hydrological

features (e.g. lakes and rivers) or in areas of cloud cover and shadow. DEM(s) created by the Polar Geospatial Center from DigitalGlobe, Inc. imagery.

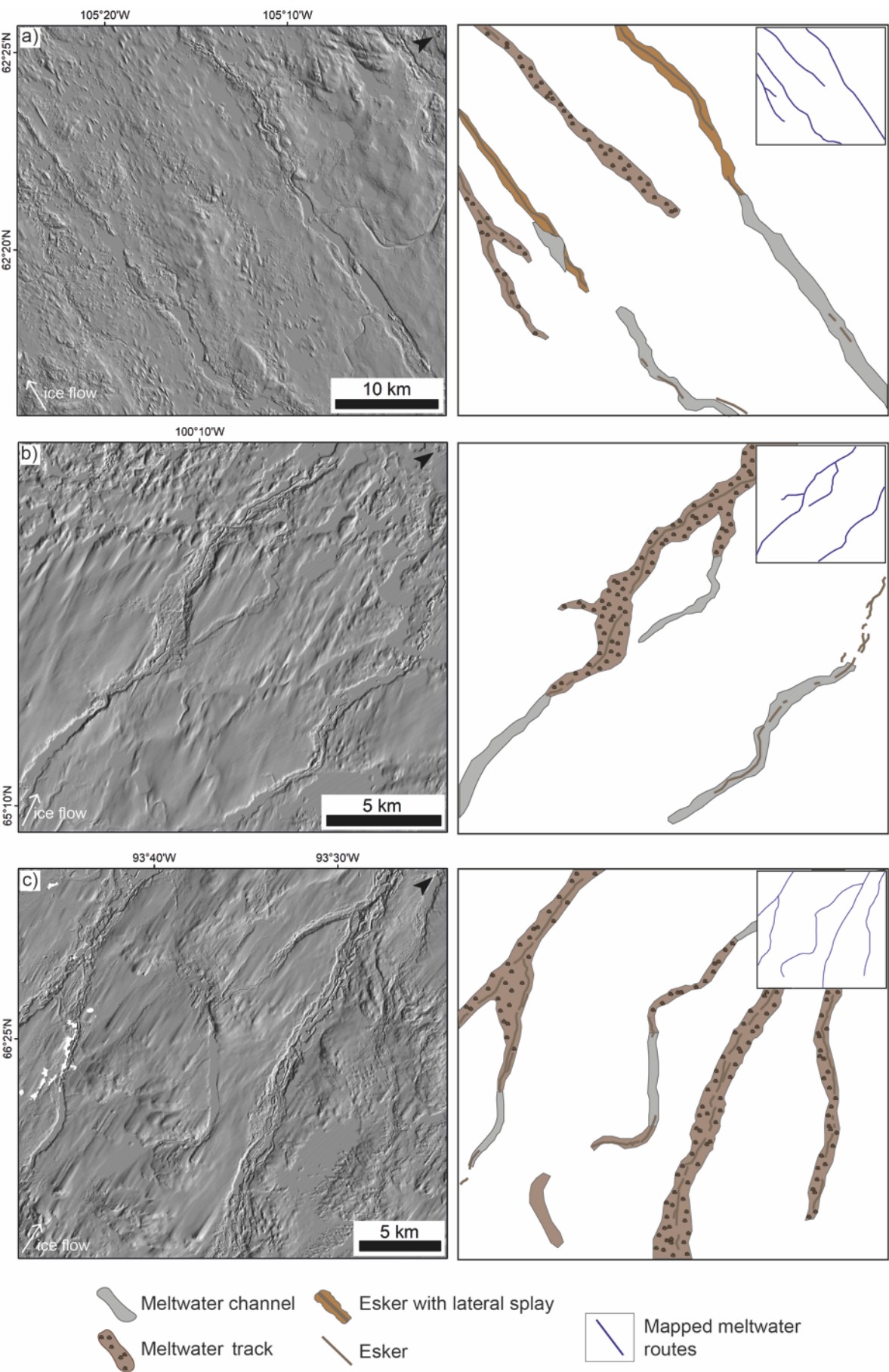

| | |
|---|---|
| Meltwater channel | Esker with lateral splay |
| Meltwater track | Esker |
| | Mapped meltwater routes |

Despite variations in expression (e.g. relief, definition and the presence or absence of hummocks, glaciofluvial material and eskers), meltwater tracks and meltwater channels are both associated with eskers (Fig. 2) and form an integrated and coherent large-scale spatial pattern (Fig. 5). Furthermore, both features have a qualitatively similar width range of several hundred meters to ~3 kilometres (Fig. 9).

However, the null hypothesis that the data in each pairing are from the same continuous distribution using the two-sample Kolmogorov-Smirnov test could not be rejected for any pairings (esker, esker with splay, meltwater channel and meltwater track) at the 5 % significance level.

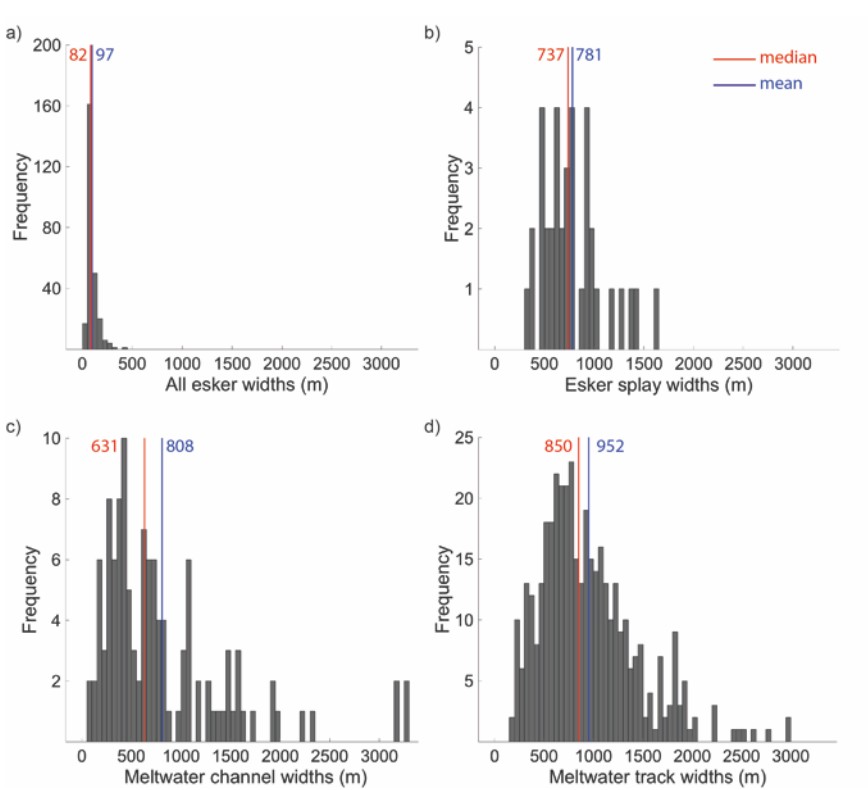


**Figure 9.** Width distributions (in metres) of (a) all esker ridges (n = 259), (b) eskers with lateral splays (n = 37), (c) meltwater channels (n = 118) and (d) meltwater tracks (n = 408) from the whole study area. The median is marked on in red and the mean in blue.


**4.4 Controls on the width and expression of meltwater landforms**

Most subglacial meltwater landforms occur within areas of till (Fig. 10). Meltwater tracks, meltwater channels and eskers with lateral splays are overrepresented in areas of till blanket, while esker ridges are strongly underrepresented. Meltwater features appear most commonly over areas of metamorphic bedrock, although meltwater channels (incisional features) are overrepresented on more erodible sedimentary rocks.

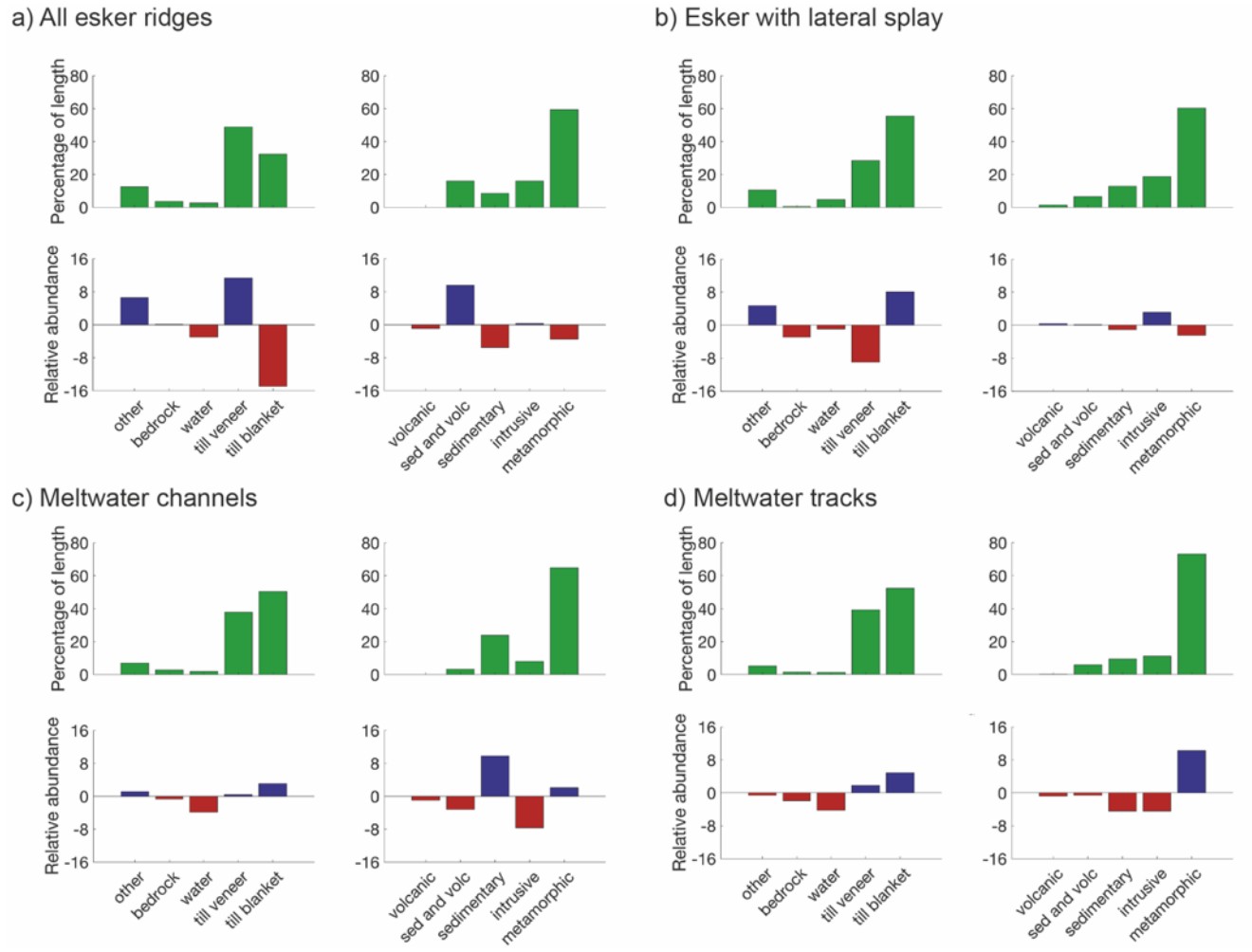

**Figure 10.** Substrate control on geomorphological expression. Occurrence (percentage of length) and relative abundance of different meltwater features over varying surface substrates (Fulton, 1995) and background geology (Wheeler et al., 1996). 'Other' includes marine, lacustrine and glaciofluvial sediments. Blue represents over representation and red represents under representation.

Figure 11 reveals high topographic variability in the NE of the study area. This coincides with the highest density of meltwater routes. Palaeo-ice streams are rare in the Keewatin District region (Stokes and Clark, 2003a, 2003b; Margold et al., 2018), but where they do occur, meltwater routes are noticeably sparser (Fig. 12). Comparing the spatial density of meltwater routes inside and outside of the ice streams (calculated simply as total length of meltwater routes per unit area) shows that the two datasets are statistically different (p = 0.03). On the bed of the Dubawnt Lake Ice Stream, meltwater routes also exhibit a more dendritic arrangement and extend further towards the ice divide.

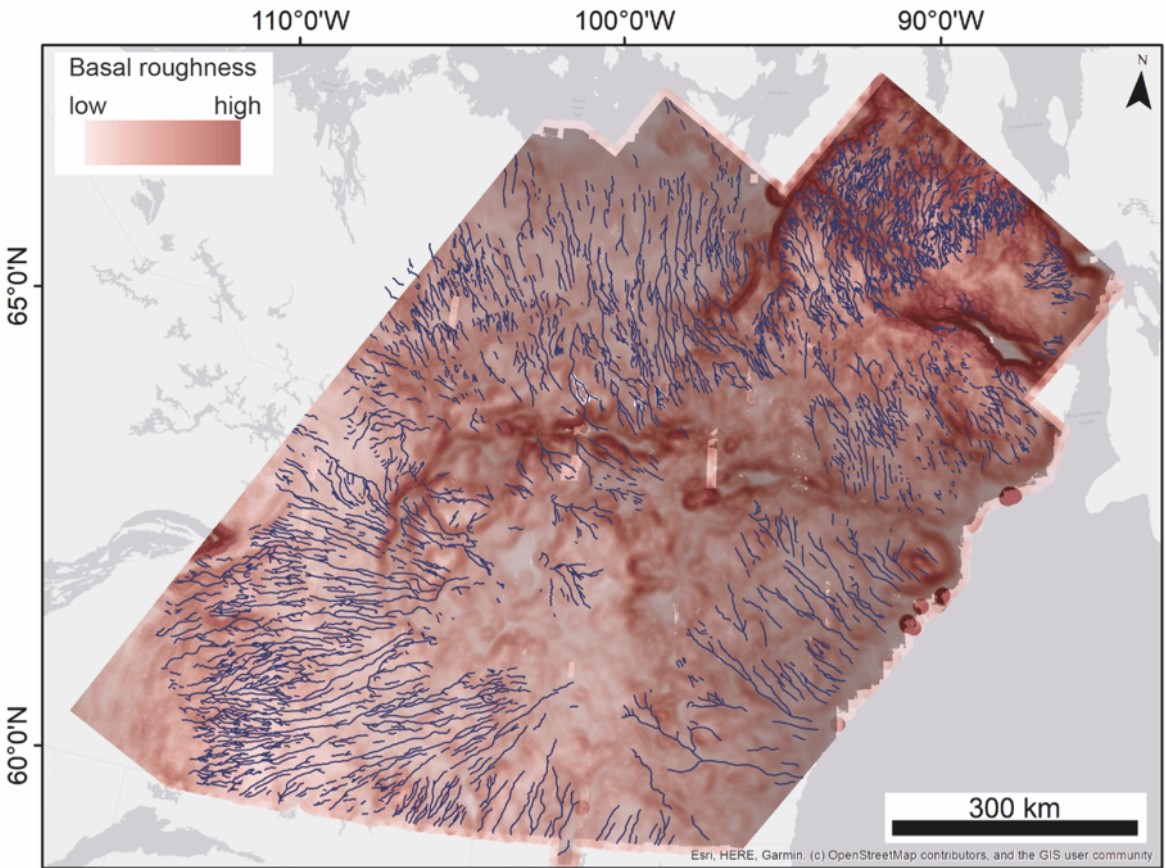

**Figure 11.** Meltwater routes overlain on local bed roughness calculated for the approximate wavelength expected to be relevant for the transfer of basal undulations to the ice surface). This is where the densest surface meltwater networks and ponding is likely to occur given sufficient melt conditions (Ignéczi et al., 2018). DEM created by the Polar Geospatial Center from DigitalGlobe, Inc. imagery.

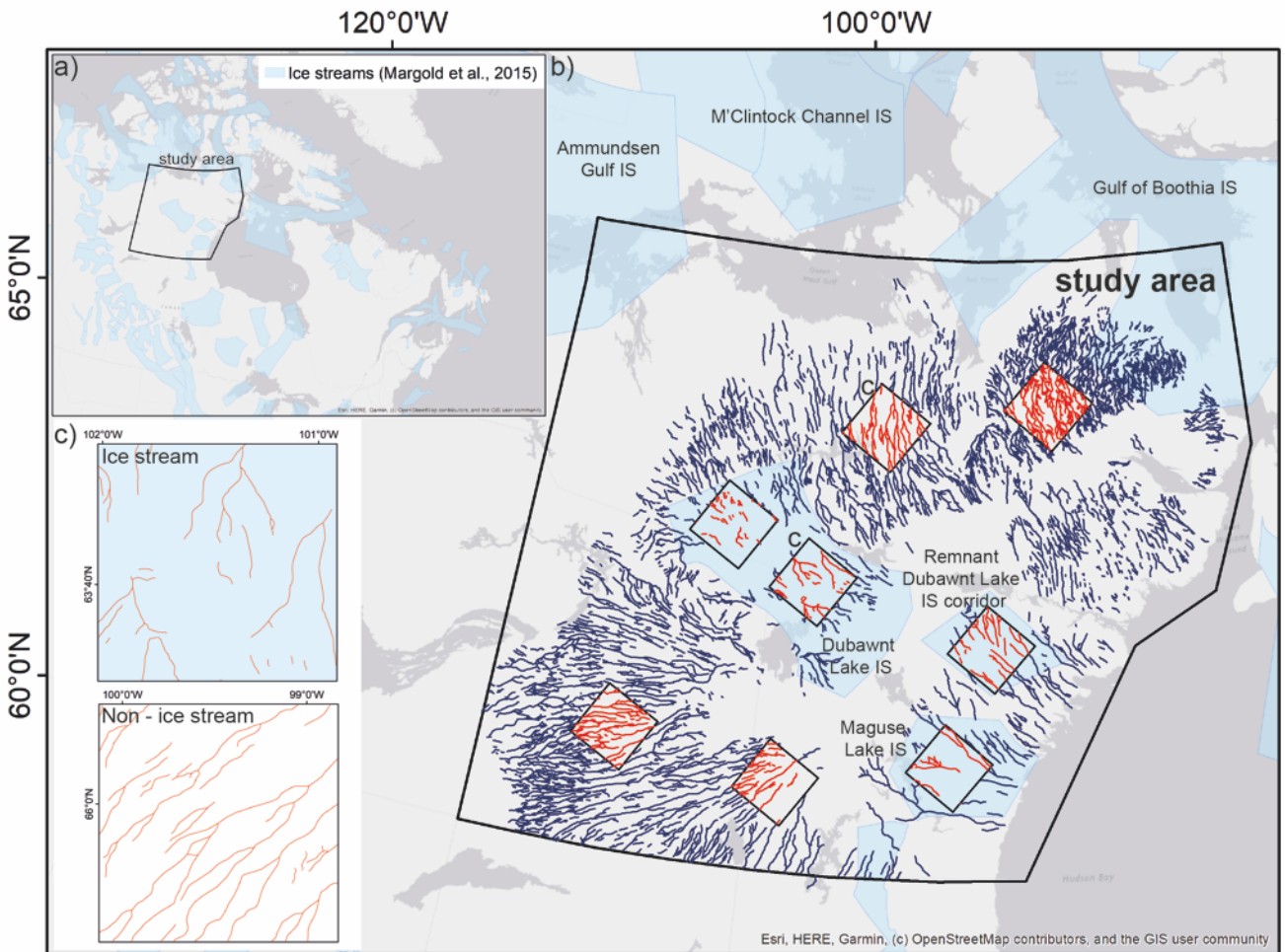

**Figure 12.** (a) Comparison of meltwater routes and palaeo-ice streams (Margold et al., 2015); (b) Spatial density was calculated for each of the randomly placed sample boxes (100 x 100 km). Ice stream density was compared to the non-ice stream density using a two sample t-test. The null hypothesis is rejected at the 5% significance level (p = 0.03); c) Visual comparison between meltwater routes in ice stream and non-ice stream areas.

To explore potential controls that govern how meltwater landform expression changes with variable background conditions (e.g. substrate, geology, topography), measurements of width, feature type and substrate were extracted along individual meltwater routes (Fig. 13). Although there is not a consistent ratio between esker width and the associated width of the meltwater track or meltwater channel when measured at the same location, there is a general positive relationship between the two, specifically when following topographic steps (e.g. Fig. 13a and 13d) and after the merging of tributaries (e.g. Fig. 13b and 13d). In Fig. 13a for example, a large increase in width (883 – 1550 m) is associated with an increase in elevation (~ 70 m over 6 km),

which also coincides with a transition from a strongly negative feature (a meltwater channel) to a positive relief depositional feature (esker with lateral splay). This sharp transition may be related to the emergence of the meltwater route out of the Thelon sedimentary basin.

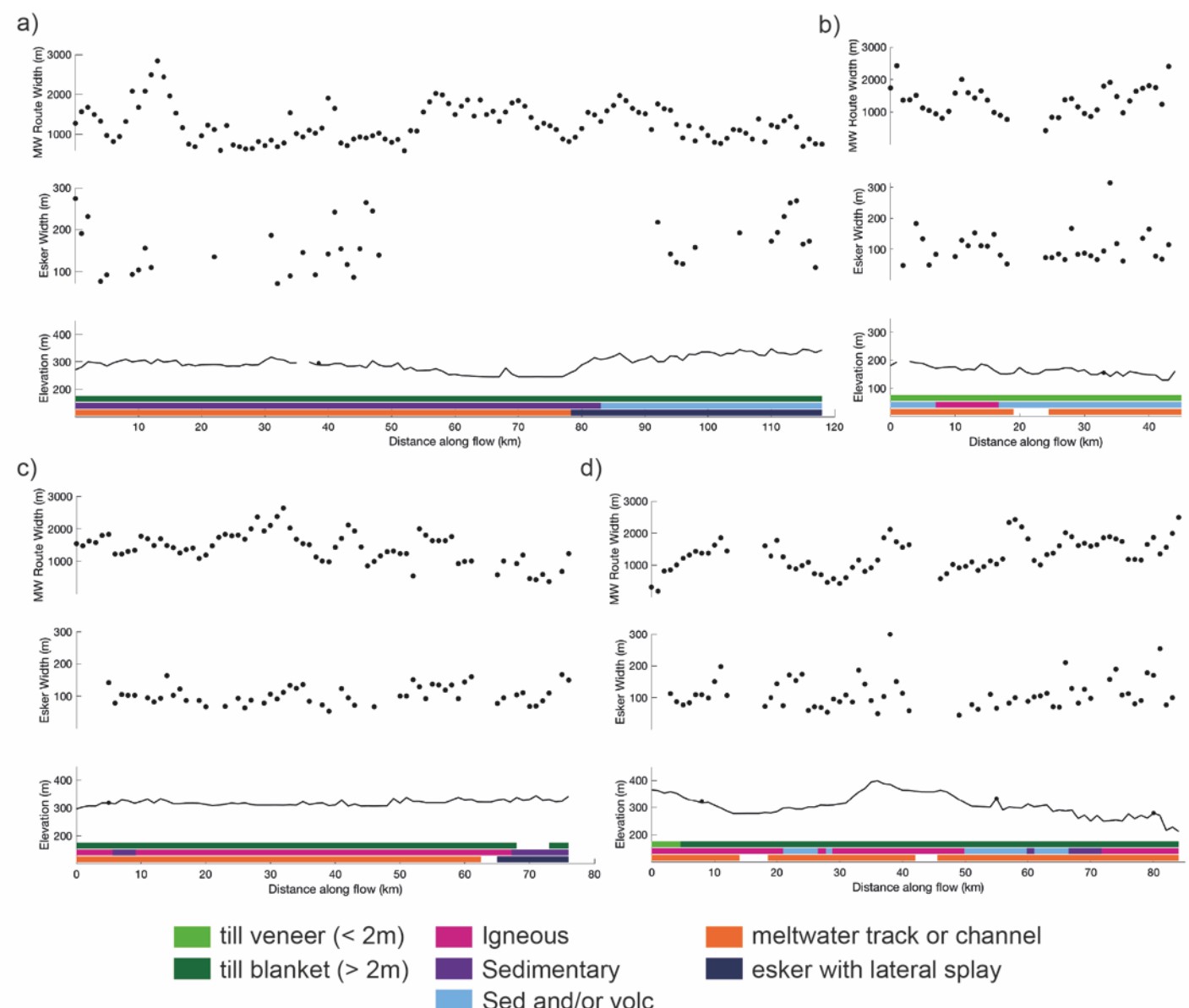

**Figure 13.** Exploring local-scale controls on meltwater route width and type. Detailed profiles (sampled at 1 km intervals along individual meltwater routes (location identified in green in fig. 5 ) show how esker width, elevation, feature expression and surface substrate vary along flow from the interior (left) to the exterior (right). Black points on the elevation plot represent the location of joining tributaries.

## 5. Discussion

Our new meltwater routes map shows that meltwater tracks and meltwater channels, which flank and connect (in an along-flow direction) esker ridges, are a dominant part of the landscape across the former Keewatin sector of the LIS. Mapping complete drainage pathways means we are better able to identify regional meltwater drainage patterns and unravel controls on feature expression.

The large-scale distribution and pattern of meltwater tracks and meltwater channels exhibit several key similarities, including width, spacing, association with eskers and occurrence within an integrated network characterised by transitions to and from different expressions along individual meltwater routes (Fig. 8). Together, this provides strong evidence that these meltwater landforms are varying expressions of the same phenomenon and we therefore group these features with widths in the order of 100s to 1000s of meters and term them meltwater corridors (Table 2). This is consistent with previous conceptual work linking meltwater landforms. For example, Sjogren et al., (2002) identify various tunnel valley (meltwater channel) expressions that they attribute to different developmental stages, from discontinuous through to fully developed valleys. Peterson and Johnson (2018) suggest that negative relief hummock corridors (meltwater tracks) are a type of tunnel valley and positive relief hummock corridors are equivalent to 'glaciofluvial corridors' in Canada (e.g. Utting et al., 2009).

Esker splays also have similar widths and a close spatial association with meltwater corridors (e.g. transitions along flow or occurring within meltwater corridors). However, it is possible that some or even all of these features were deposited marginally (e.g. Hebrand and Amark, 1989) rather than in subglacial cavities. In fact, marginal deposition is supported by the fact that some of the esker splays align across flow in line with estimated ice sheet isochrones (Dyke et al., 2003). Nonetheless, it is difficult to constrain their formation from geomorphology alone.

While recognised previously in local case studies (e.g. St-Onge, 1984; Rampton, 2000; Utting et al., 2009), we confirm that across this 1 million km$^2$ area of the former LIS, meltwater corridors of varying geomorphic expression are widespread

(captured at 84 % of all sample points) rather than an isolated phenomenon. Esker ridges are captured at just 43 % of sample locations, however, we do note that the presence or absence of an esker at the sample point may not be indicative of the entire length of the meltwater route as in many cases the esker ridges within a meltwater corridor are fragmented. Nonetheless, we suggest that the model of R-channels

across the Canadian Shield (e.g. Clark and Walder, 1994) is an oversimplification and may under-represent the modes and thus coverage of drainage in this sector and fail to capture important processes recorded on the bed.

Holistic mapping of meltwater routes including features cut up into the ice (i.e.

eskers) and features cut down into the bed (i.e. meltwater corridors) creates a more complete and less fragmented drainage map than mapping individual features (Fig. 7). The broad scale pattern of palaeo-drainage radiating out from the former Keewatin Ice Divide, which remains noticeably absent of meltwater evidence (Fig. 5), is consistent with previous studies (Shilts et al., 1987; Aylsworth and Shilts, 1989; Storrar

et al., 2013, 2014a), but our mapping results in a greater density, narrower spacing (Fig. 6) and higher number of tributaries.

 **Table 2.** Proposed classification for subglacial meltwater traces observed on palaeo-ice sheet beds. Meltwater routes encompass all visible evidence and consists of negative and negligible relief meltwater corridors with widths in the order 100's of meters and esker ridges with widths in the order of 10's of meters.

| Proposed Classification | Description | Example |
|---|---|---|
| Meltwater route | All visible traces of subglacial meltwater drainage (i.e. all of below) |  |
| Meltwater corridor | - Meltwater channel<br>- Tunnel channel<br>- Tunnel valley<br><br>- Hummock corridor (negative) (e.g. Peterson and Johnson, 2018; Lewington et al., 2019)<br>- Erosional corridor (e.g. Burke et al., 2011)<br>- Esker corridor (e.g. Sharpe et al., 2017)<br>- Meltwater corridor (e.g. Rampton, 2000)<br>- Washed zone (e.g. Ward et al., 1997)<br><br>- Hummock corridor (positive) (e.g. Peterson and Johnson) |  |
| Esker with lateral splay | Esker ridge flanked by lateral splay (e.g. Cummings et al., 2011) |  |
| Esker ridge | Single, multiple or anastomosing esker ridges |  |

### 5.1 Proposed model for meltwater corridor formation


To interpret palaeo-landforms and reconstruct subglacial meltwater behaviour an understanding of the processes that formed the landforms is needed. This is the 'glacial inversion' problem (e.g. Kleman and Borgström, 1996). One approach to understanding glacial processes is through contemporary observations. In this

section, we demonstrate how contemporary observations and modelling of the hydraulically connected distributed drainage system (e.g. Hubbard et al., 1995; Bartholomaus et al., 2008; Andrews et al., 2014; Hoffman et al., 2016) is consistent with the form and distribution of mapped meltwater corridors and can explain the range of depositional to erosional signatures observed in the study area.


Although hydrological theory dictates that a conduit in steady state will operate at lower pressure than the surrounding distributed system, large or relatively rapid fluctuations in surface meltwater inputs (compared to the rate at which conduits expand from melting caused by turbulent heat dissipation) during the melt season

mean the system is rarely in steady-state (Bartholomew et al., 2012). Once a conduit system has evolved gradually to accommodate high meltwater fluxes (Cowton et al., 2013), it is likely to operate at lower pressure than the surrounding high-pressure weakly connected system during periods of low meltwater input (e.g. at night and later in the melt season), thus drawing water in (Fig. 14a and 14c). During this phase, the

geomorphic work in the hydraulically connected distributed drainage system is likely limited by the small cross-sectional area of passage and slow water movement (Willis et al., 1990; Alley et al., 1997). However, there could be migration of finer sediments into the central conduit contributing to gradual lateral channel growth over time; this has been invoked to explain steady state growth of tunnel valleys for example, (e.g.

Boulton and Hindmarsh, 1987).

Variations in borehole water pressure measurements observed at glaciers in the Alps (e.g. Hubbard et al., 1995; Gordon et al., 1996), Canada (e.g. Rada and Schoof, 2018) and Alaska (e.g. Bartholomaus et al., 2008), ice velocity measurements

taken from the Greenland Ice Sheet (e.g. Tedstone et al., 2014) and numerical modelling (e.g. Werder et al., 2013), suggest that large or rapid meltwater inputs can cause spikes in conduit water pressure (Cowton et al., 2013). This temporarily

reverses the hydraulic potential gradient and causes water to flow out of the conduit and into the surrounding hydraulically connected distributed drainage system (Fig. 14b).

The width of the hydraulically connected distributed drainage system affected and the form the drainage takes likely depends on the magnitude of the pressure perturbation, determined by the volume and rate of meltwater input, basal substrate and antecedent conduit conditions (e.g. Iken and Bindschadler, 1986; Andrews et al., 2014; Rada and Schoof, 2018; Nanni et al., 2020). For example, the hydraulically connected distributed drainage system is widest during the early melt season when the hydrological system is less developed and the system can be easily over pressurised. Later during the summer, the same magnitude meltwater input does not cause the same degree of over pressurisation as conduits have increased their capacity to accommodate fluctuations in surface meltwater inputs (e.g. Rada and Schoof). The magnitude of the pressure perturbation is also likely to result in different forms of drainage through the hydraulically connected distributed drainage system. This may range from expansion of linked cavities during smaller magnitude events (Fig. 14b1), to drainage re-organisation into braided canals (e.g. Catania and Paola, 2001) or anastomosing conduits (e.g. Gulley et al., 2012) (Fig. 14b2), and finally to narrow sheet floods (e.g. Russell et al., 2007) (Fig. 14b3). While water flows laterally out of the conduit down the pressure gradient during these high-pressure events, the dominant flow direction is still parallel to the main conduit (i.e. down-flow). Fluctuations in pressure within the subglacial conduits may therefore be key to understanding how sediment is accessed and eroded and for explaining variations in sediment flux.

The hydrological system is responsible for transporting the majority of subglacial sediment (e.g. Walder and Fowler, 1994; Richards and Moore, 2003). This is influenced by access to sediment (e.g. Willis et al., 1996; Burke et al., 2015) and subglacial water velocity (e.g. Walder and Fowler, 1994; Ng, 2000). Water flow through the distributed system is slow and inefficient with limited sediment mobilisation and restricted transport (e.g. Willis et al., 1990; Alley et al., 1997). Faster and more turbulent water flow within conduits is more efficient at eroding and transporting sediment and this capability increases rapidly with increased discharge (Alley et al., 1997). However, conduits cover only a small fraction of the bed, which restricts their

ability to erode and transport sediment across large areas (Alley et al., 2019). Thus, there is a need for an additional mechanism(s) to access surrounding sediments. While deformation of till into channels (e.g. Boulton and Hindmarsh, 1987) and lateral conduit migration (e.g. Beaud et al., 2018b) have been proposed, our model focusses on the connection of the hydraulically connected distributed drainage system to the conduit in a range of forms (Fig. 14b). This idea is grounded in the wider glacio-fluvial literature, which suggests that rapid increases in water input create high water pressures that overwhelm the conduit and surrounding drainage system, causing both increased access to and high enough water velocities to carry sediment (e.g. Swift et al., 2002, 2005b; Gimbert et al., 2016; Delaney et al., 2018). This enhanced sediment transport typically occurs at the start of the melt season (e.g. Liestol, 1967; Hooke et al., 1985; Collins, 1989; 1990), but also during large meltwater events (e.g. precipitation (Delaney et al., 2018)). An extreme example is during the 1996 Icelandic jökulhlaup, when a subglacial flood evacuated sediment creating a large tunnel valley (Russell et al., 2007). Thus, fluctuations in subglacial conduit pressure within the ablation zone of ice sheets are likely to be a key mechanism by which sediment either side of the conduit is accessed and mobilised.

In our proposed model, meltwater corridor relief is caused by localised turbulent flow enhancing erosion (e.g. Rampton, 2000). Sedimentological evidence suggests that hummocks within the corridors occur as a result of both erosional and depositional processes. Our proposed model can account for either process, with hummocks forming as a result of: (i) erosion by high energy turbulent water flow along conduits and across the hydraulically connected distributed system (e.g. Rampton, 2000; Peterson et al., 2018); or (ii) deposition during waning stages of the flood within cavities either melted up into the overlying ice by turbulent floods (e.g. Utting et al., 2009), or minor conduits and linked cavities alongside the conduit (e.g. Brennand, 1994). Hummocks may also form as a combination of processes akin to the interpretation of triangular shaped landforms ('murtoos'), which are attributed to subglacial till transported by creep and subsequently eroded and shaped by subglacial meltwater (Mäkinen et al., 2017; Ojala et al., 2019).

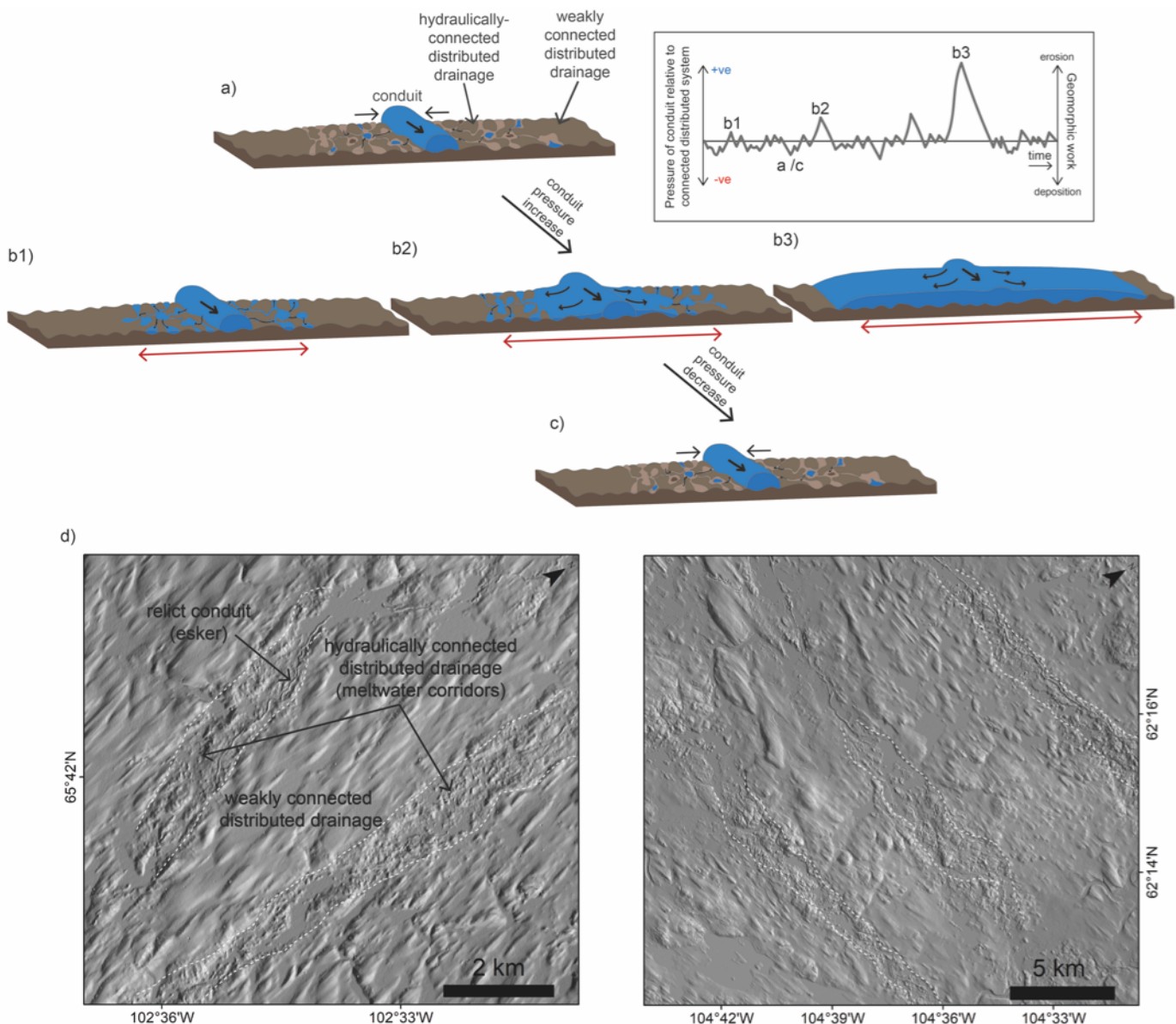

**Figure 14.** Effects of pressure perturbations on the hydraulic conductivity within the conduit connected distributed subsystem: (a) steady state – water is drawn in from across the connected distributed system into the conduit (low pressure) down the pressure gradient, geomorphic work is limited to the conduit, although there may be some lateral sapping (e.g. Boulton and Hindmarsh, 1987); (b) During over-pressurisation events water is forced out of
the conduit across and into the surrounding hydraulically connected distributed drainage system. The width and form (i.e. flood or cavity expansion) this takes likely depends on the magnitude of the pressure perturbation. Geomorphic work (erosion and deposition) likely occurs during this phase; (c) Return to steady state as meltwater input decreases or the conduit expands to accommodate a sustained increase in input; (d) Proposed cumulative
geomorphic imprint of the process over time, creating the meltwater corridors (white dashed lines) preserved on the landscape today. The inset in the upper right corner demonstrates that

pressure perturbations within the conduit fluctuate throughout the melt season and vary in size from regular diurnal fluctuations (e.g. b1) to irregular larger 'events' (e.g. b2 and b3), which may represent precipitation or supra/sub-glacial lake drainages.

In areas of thicker sediment, pressure-driven drainage reorganisation, which takes the form of cavity expansion or sheet floods in other areas, may result in braiding across the hydraulically connected distributed system (e.g. Catania and Paola, 2001). This is consistent with braided meltwater channels identified within tunnel valleys in the North Sea (Kirkham et al., 2020), while the spacing and shape of the hummocky topography observed along meltwater corridors has been interpreted as remnants of braided conduits and intervening bars (e.g. Dahlgren, 2013; Peterson et al., 2018). Likewise water driven from the conduit into the hydraulically connected distributed system during discrete recharge from moulins has been recognised to form anastomosing conduits (Gulley et al., 2012). Thus, anastomosing or braided conduits moving around at the bed and formed during conduit over pressurisation may produce an erosional signature wider than the individual conduit.

While we fully expect to see transient conduit migration and reconfiguration during conduit over pressurisation, we also do not rule out the possibility that individual conduits could migrate laterally across the bed over longer periods of time, for instance due to changes in ice thickness, subglacial topography, and regional and local basal water pressure. This theory has been invoked to explain the formation of some tunnel valleys by the lateral merging of a series of smaller discrete drainage events over time (e.g. Jørgensen and Sandersen, 2006; Kehew et al., 2012; Beaud et al., 2018b). Indeed, seismic tremor observations suggest that areas with low hydraulic gradients (i.e. flatter parts of the bed, higher up the ice sheet) are characterised by quasi-stable conduit configurations where water is less restricted and can flow through multiple conduits, which alternate and migrate on multiday timescales (Vore et al., 2019). In contrast, the same research suggests that nearer the margin where the hydraulic gradient is steeper, conduits are relatively stable in space (Vore et al., 2019). This fits with esker sinuosity studies which demonstrate that eskers are often very straight (median sinuosity 1.04 on the Canadian Shield) with esker segments aligning over distances of 10s of km's (Storrar et al., 2014a). If the conduit migrated extensively in

the marginal zone, we would expect to find more sinuous eskers or esker sections which are offset. Our work and earlier studies indicate that esker ridges can be superimposed on hummocks within meltwater corridors, but to date, there are no examples of hummocks overlying eskers (e.g. Peterson et al., 2018). Together, this suggests that the formation of eskers are separated in time from the meltwater corridors in which they often occur (e.g. Beaud et al., 2018a; Hewitt and Creyts, 2019; Livingstone et al., 2020). This supports the notion that palaeo-ice sheet beds are a composite picture of geomorphic effects, combining different stages and potentially different subglacial drainage regimes (Greenwood et al., 2016).

Using an inland limit of 60 km for subglacial channelisation (e.g. Chandler et al., 2013), and minimum and maximum retreat rates in the study area (~ 230 m yr$^{-1}$ - ~ 540 m yr$^{-1}$), we estimate the time likely spent beneath the channelised zone influenced by surface meltwater inputs at between ~ 110 and ~ 260 years. We therefore suggest that meltwater corridors reflect the geomorphic work arising from repeated pressure perturbations in the ablation zone over 10s – 100s years. The most significant erosion likely occurred where fluctuating surface meltwater inputs were clustered (e.g. Alley et al., 2019) or where cumulative upstream drainage produced the threshold shear stresses required to erode and transport the substrate, which may have occurred upstream of the peak local meltwater input. While the location of surface meltwater drainage and discrete water input points (crevasses and moulins) are important controls on the distribution of subglacial drainage at the bed (e.g. Decaux et al., 2019), observations suggest that both supraglacial networks (e.g. Koziol et al., 2017) and moulin locations (e.g. Catania and Neumann, 2010) are relatively stable, at least over decadal timescales. Where changes in surface meltwater input areas are observed, this occurs over relatively short distances (~ 300 m$^2$) with the new routes likely occurring along the same drainage axes and thus not resulting in significant subglacial drainage system reorganisation (Decaux et al., 2019). This is consistent with geomorphological evidence, which reveals a coherent drainage network (Fig. 5) with individual meltwater corridors extending 100s km (Table 1).

The variable extent to which the hydraulically connected drainage system (and thus the meltwater corridors width) is affected by conduit over pressurisation may be influenced by ambient variations in the conduits lateral hydropotential gradient (e.g.

narrower meltwater corridors within a steep 'hydropotential valley'). However, we
suggest the key control will be the magnitude of the pressure perturbation, which will
vary depending on meltwater input and antecedent subglacial drainage conditions. If
a corridor represents a single maximum flow, meltwater corridor widths in this study
(0.05 – 3.3 km (mean 0.9 km)) are comparable to measurements in alpine settings (~
140 m (Hubbard et al., 1995; Gordon et al., 1998)) and modelled ice sheet settings (~
2 km (Werder et al., 2013)).

## 5.3 Exploring potential controls on network patterns and variations in expression of meltwater routes

In this section, we explore spatial controls governing the overall pattern of the
subglacial hydrologic network, as well as variations in meltwater landform expression
(i.e. the patterns of and balance between erosion and deposition and the resulting
geomorphic expression) along individual meltwater routes. Erosional and depositional
features are frequently observed along the same meltwater route and even at the
same location, for example eskers with lateral splays occurring within meltwater
corridors.

There is a high degree of channelisation across the Keewatin sector of the ice
sheet bed, but channelisation is not uniform and the densest areas of meltwater routes
coincide with the 'roughest' basal topography (Fig. 11). This may be the result of
subglacial drainage route fragmentation around bed obstacles, with a greater number
of tributaries and broken patterns common in regions of high bed roughness (e.g. Test
Site 3). Basal topography also preconditions the large-scale spatial structure of
surface drainage (Ignéczi et al., 2018) and the association between rough areas and
dense clusters of meltwater routes could be a response to more surface water
penetrating to the bed as the result of extensive crevassing. For a typical melt season
in west Greenland, crevasses capture a significant amount of surface water – more
than moulins or the hydrofracture of surface lakes (Koziol et al., 2017). Surface
meltwater inputs are thought to be an important control on the distribution of drainage
across the bed (e.g. Gulley et al., 2012; Banwell et al., 2016) and the formation and
evolution of subglacial meltwater landforms (e.g. Banerjee and McDonald, 1975; St-

Onge, 1984; Hooke and Fastook, 2007; Storrar et al., 2014b; Livingstone et al., 2015; Peterson and Johnson, 2017).

There are significantly fewer meltwater routes coinciding with palaeo-ice stream locations – particularly the Dubawnt Lake Ice Stream (Fig. 12). In addition, the network pattern of meltwater routes corresponding with the location of the Dubawnt Lake Ice Stream are more dendritic and extend further towards the ice divide. These observations are consistent with Livingstone et al., (2015), who find fewer eskers on palaeo-ice stream beds where modelled subglacial meltwater drainage is greatest. We suggest the scarcity of meltwater routes beneath palaeo-ice streams could be the result of (i) lower ice-surface slopes and hydraulic potential gradients, which favour distributed rather than channelised drainage (e.g. Kamb, 1987; Bell, 2008) or (ii) lack of preservation beneath fast flowing ice (Boulton, 1996). Where channelised drainage does occur beneath palaeo-ice streams, networks are typically more dendritic, which may also be the result of shallower hydraulic gradients and lower relief bed topography enabling greater lateral water flow.

Dynamic ice mass loss via streaming or surging (and subsequent melting and iceberg calving) has implications for ice sheet stability (e.g. Bell, 2008; Christianson et al., 2014; Christoffersen et al., 2014). The Keewatin sector of the LIS had a relatively low spatial density of ice streams compared to the western and southern margins (Margold et al., 2015; Stokes et al., 2016). This may be partially attributed to the low relief, resistant bed of the shield which was unable to provide the fine grained sediments required to lubricate ice flow and the fact that the margin reached this position later during deglaciation when the remaining ice sheet was much smaller (e.g. Margold et al., 2015; Stokes et al., 2016). Nonetheless, we also suggest that efficient evacuation of meltwater through the dense channelised network, which developed in this region during the final stages of deglaciation, as the climate warmed (Storrar et al., 2014b), could have inhibited the development of fast flow and potentially contributed to the shut-down of existing ice streams. This is consistent with recent physical modelling (Lelandais et al., 2018) and modern temporal observations that link decadal-scale ice-flow decelerations with more pervasive and efficient drainage channelisation driven by increased surface meltwater inputs to the bed (Sole et al., 2013; Tedstone et al., 2014; van de Wal et al., 2015; Davison et al., 2019) and vice

versa (Williams et al., 2020). If this hypothesis is correct we would expect to see this large-scale inverse spatial relationship between channelisation and ice streaming in other palaeo-ice sheet settings. This potential drainage control on ice-sheet velocity and stability may also influence the pace of deglaciation; we note slower retreat rates (~230 m yr$^{-1}$) in the northwest of the study area, which coincide with the highest density of meltwater routes, compared to much faster retreat rates (~540 m yr$^{-1}$) associated with the sparsest meltwater routes. This conclusion is tentative given uncertainty in the regions deglacial chronology (Dyke et al., 2003) and the many other factors that can influence retreat rate, and thus requires further testing.

At a large-scale, there is a general tendency for meltwater routes to preferentially form on till, which is more easily eroded than bedrock and where geomorphic evidence is likely to be better developed. Eskers are over-represented on harder, more resistant rock (Fig. 10d) where R-channels are more likely to form (Clark and Walder, 1994; Storrar, 2014a), while there is a slight tendency for meltwater channels (i.e. incisional features) to form on the softer, more erodible sedimentary rock (Fig. 10b). Eskers with lateral splays (i.e. depositional features) appear preferentially on till blankets (Fig. 10c) where there is an abundance of sediment that may overwhelm and clog up the conduit (e.g. Burke et al., 2015), while isolated esker ridges favour thin till and are under-represented on thick till. Though detailed long-profiles (Fig. 13) hint at local relationships between bed substrate changes and the resultant landform expression, we caution against the assumption that this is a widespread occurrence rather than an isolated coincidence.

### 5.3 Implications

Western sectors of the contemporary Greenland Ice Sheet are broadly analogous to our study area: both are underlain by resistant Precambrian Shield rocks and both experience(d) rapid retreat and high meltwater production rates. This is also similar to southern Sweden, which lay beneath the palaeo Scandinavian Ice Sheet, where similar geomorphic features to those described here, occur extensively (e.g. Peterson et al., 2017; Peterson and Johnson, 2017). This study therefore has potential implications for our understanding of the impact of subglacial hydrology on overlying ice dynamics and ice flow regulation of past, current and future ice sheets.

The interaction between a subglacial conduit and the surrounding hydraulically connected distributed drainage system is believed to be widespread in contemporary glaciological settings (e.g. Hubbard et al., 1995; Gordon et al., 1996; Bartholomaus et al., 2008; Werder et al., 2013; Tedstone et al., 2014) and has been identified as key to understanding ice velocity variations and predicting future ice sheet mass loss (Davison et al., 2019). However, the true extent and influence of the hydraulically connected distributed drainage system beneath the Greenland Ice Sheet is unknown due to the challenge of observing contemporary subglacial environments. Palaeo-studies, such as this one, offer the potential to reveal new insights into the nature and configuration of the subglacial hydrological system at an ice sheet scale and potential quantification of how much of the bed and ice surface dynamics were affected by subglacial meltwater.

Based on our proposed model, we estimate the coverage of each drainage element across the bed of the Keewatin Ice Sheet. Conduits (i.e. eskers) cover ~ 0.5 % of the bed based on an average esker width of 100 m and spacing of 18.8 km (Storrar et al., 2014a). The coverage of conduits and the surrounding hydraulically connected distributed drainage system (i.e. meltwater corridors) increases to an average of ~ 13 % using the average width and spacing of meltwater routes in this study, but could realistically vary between 5 % (lower quartile width and upper quartile spacing) and 36 % (upper quartile width and lower quartile spacing). This represents an area 25 times greater than the conduits (eskers) alone, but assumes that all meltwater routes were active at the same time.

Based on the above and while we propose a significant increase in the area of the bed influenced by surface meltwater inputs, these findings also fit with the hypothesis that the weakly-connected distributed system covers a large percentage of the subglacial bed (Hoffman et al., 2016). Our results suggest that somewhere between 64 – 95 % of the bed existed within the weakly-connected distributed system where there are no visible traces of subglacial meltwater flow. This finding is similar to Hodge (1979) who suggested that 90 % of the bed at the South Cascade Glacier in Washington was hydraulically isolated. Quantifying the relative coverage of the inactive hydraulically isolated regions of the bed and better understanding how they

regulate the active drainage regions and modulate basal traction is likely to be important for understanding ice sheet dynamics (Hoffman et al., 2016).

In contemporary settings, the hydraulically connected distributed drainage system is strongly linked to surface meltwater inputs and conduit over pressurisation. The LIS 950 is expected to have exhibited strong surface melting during the period of retreat over this area (estimated at -0.85 m yr$^{-1}$ for 9 ka) with surface ablation accounting for much of this (Carlson et al., 2009). The widespread presence of meltwater corridors across Keewatin thus complements their interpretation and reveals a geomorphic signature of this interaction.


Finally, there are large uncertainties as to how sediment is accessed by subglacial meltwater and transported to conduits (Alley et al., 2019). We suggest that the over pressurisation of conduits and their interaction with the surrounding hydraulically connected distributed drainage is a key driver of sediment erosion and entrainment 960 within the ablation zone and may help address this question. As a result, conduits may be less sediment limited than previously thought and, much like the evolution of the subglacial drainage system (e.g. Schoof, 2010), rates of subglacial fluvial erosion may be strongly controlled by melt supply variability rather than the overall input of meltwater into the system.


## 6. Conclusions

We used the ArcticDEM to identify and map all visible traces of subglacial 970 meltwater drainage in the Keewatin sector of the former LIS. We found that wider meltwater features (meltwater tracks and meltwater channels) on the order of 100's to 1000's m flanking or joining up intervening segments of esker ridges were common. These have previously been termed and described as different features. However, as they form part of the same integrated network and display similarities in spacing and 975 morphometry, we propose collectively grouping these features under the term meltwater corridor (Table 2). Combing esker ridges and all varying geomorphic expressions of meltwater corridors within a single meltwater routes map, we have created the first large-scale holistic map of subglacial meltwater drainage for this area.

Based on our observations and modern analogues, we propose a new model, which accounts for the formation and geomorphic variations of meltwater corridors. In this model, we propose that a principal conduit (i.e. the esker) interacts with the surrounding hydraulically connected distributed drainage network (i.e. the meltwater corridor) with the extent and intensity of this interaction, determined by the magnitude

of water pressure fluctuations within the conduit. The geomorphic expression (i.e. net erosion or deposition), is likely governed by a combination of glaciological (i.e. relative water pressure fluctuation) and background controls (i.e. topography, basal substrate and geology). Eskers likely represent the final depositional imprint of channelised drainage within the large-scale meltwater routes network close to the ice margin, while

meltwater corridors represent a composite imprint of drainage formed over 10s - 100s years. If our model is correct, the drainage footprint of the hydraulically connected distributed drainage system in this sector is 25 times greater than previously assumed from eskers alone, which only account for the central conduit.

Our results suggest that the overall distribution and pattern of drainage is influenced by background topography, with greater relief resulting in denser channelised networks, possibly due to fragmentation of subglacial drainage around basal obstacles and the result of more spatially distributed meltwater delivery to the bed. Channelised drainage is relatively rare beneath palaeo ice streams, which

instead favour distributed drainage configurations due to the lower ice surface slopes and subglacial hydraulic gradients, and likely also exhibit reduced landform preservation potential. The style of meltwater drainage may influence ice dynamics, with the high degree of channelisation observed in the region able to efficiently dewater the bed leading to slower ice-flow and limited ice stream activity.


        Finally, our results suggest that conduit over pressurisation events and the subsequent connection between conduits and the surrounding hydraulically connected distributed drainage system may be important for understanding how sediment is accessed and entrained at the bed. While conduits (eskers) alone cover

~ 0.5 % of the bed, the connected distributed drainage system (meltwater corridors), cover 5 – 36 % of the bed, providing a greater area for sediment erosion and likely the high velocity flows required to do so.

Further research should focus on determining how common the proposed
interaction between conduits and the surrounding distributed drainage system is
beneath other palaeo and contemporary ice sheets and the controls governing its
variability. We hypothesise that where less surface meltwater is delivered to the bed
or ice-surface slopes are shallower for instance when the LIS was larger and the
climate colder, the geomorphic expression will be less extensive and subtler. This is
because conduits are less likely to evolve due to lower hydraulic gradients, and their
interaction with the surrounding distributed system is limited because of invariant melt
supply. Understanding where this interaction and signature occurs will help confirm or
refute our proposed model, and develop understanding of how meltwater drainage
evolves and influences ice dynamics and mass balance over long time-scales.

**Data Availability**

The mapping of meltwater routes is archived at xxxx.

**Author Contributions**

ELML carried out the mapping and analysis with input and interpretation from co-
authors. ELML prepared the manuscript with contributions from all co-authors.

**Competing Interests**

The authors declare that they have no conflicts of interest.

**7.  Acknowledgements**

This work was funded through: "Adapting to the Challenges of a Changing
Environment" (ACCE); a NERC funded doctoral training partnership ACCE DTP
(NE/L002450/1). This work also benefitted from the PALGLAC team of researchers
who received funding from the European Research Council (ERC) under the
European Union's Horizon 2020 research and innovation programme (Grant
agreement No. 787263). DEMs were provided by the Polar Geospatial centre under
NSF-OPP awards 1043681, 1559691, and 1542736. We would like to thank an
anonymous reviewer and Flavien Beaud for their valuable comments which have

significantly improved the manuscript and have helped us to clarify our proposed model.

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

## 9. Appendices

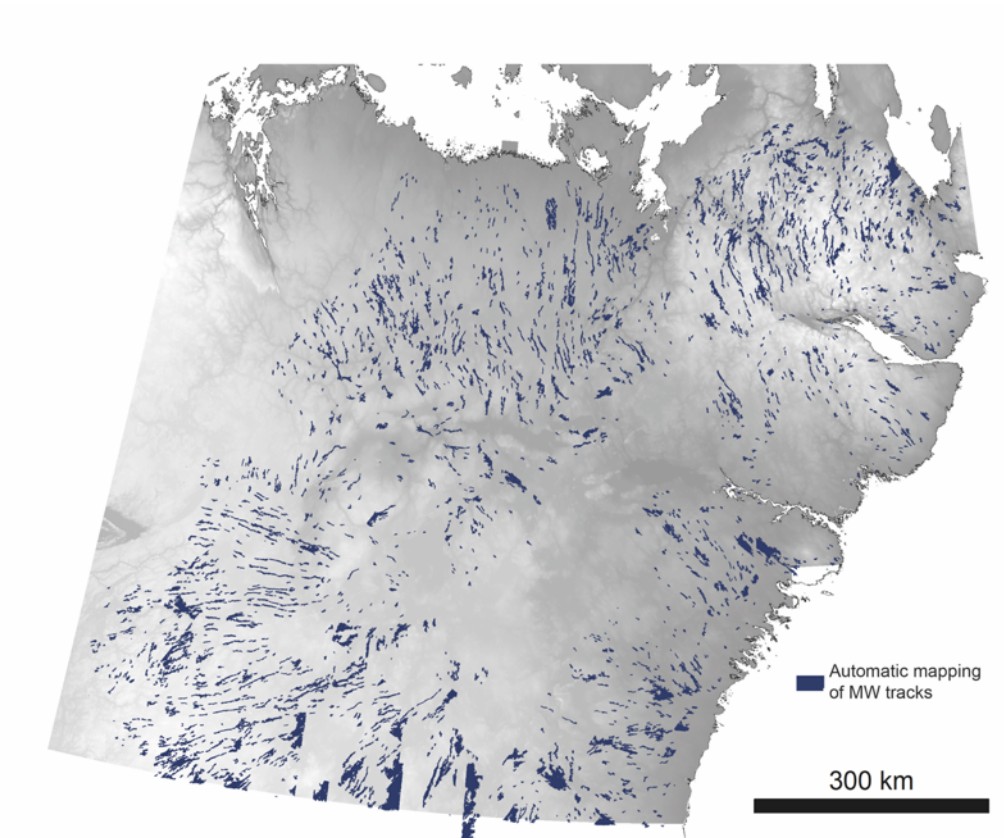

**Figure A1**. Automatic mapping output (cleaned up) for test site using code associated with
Lewington et al., 2019.

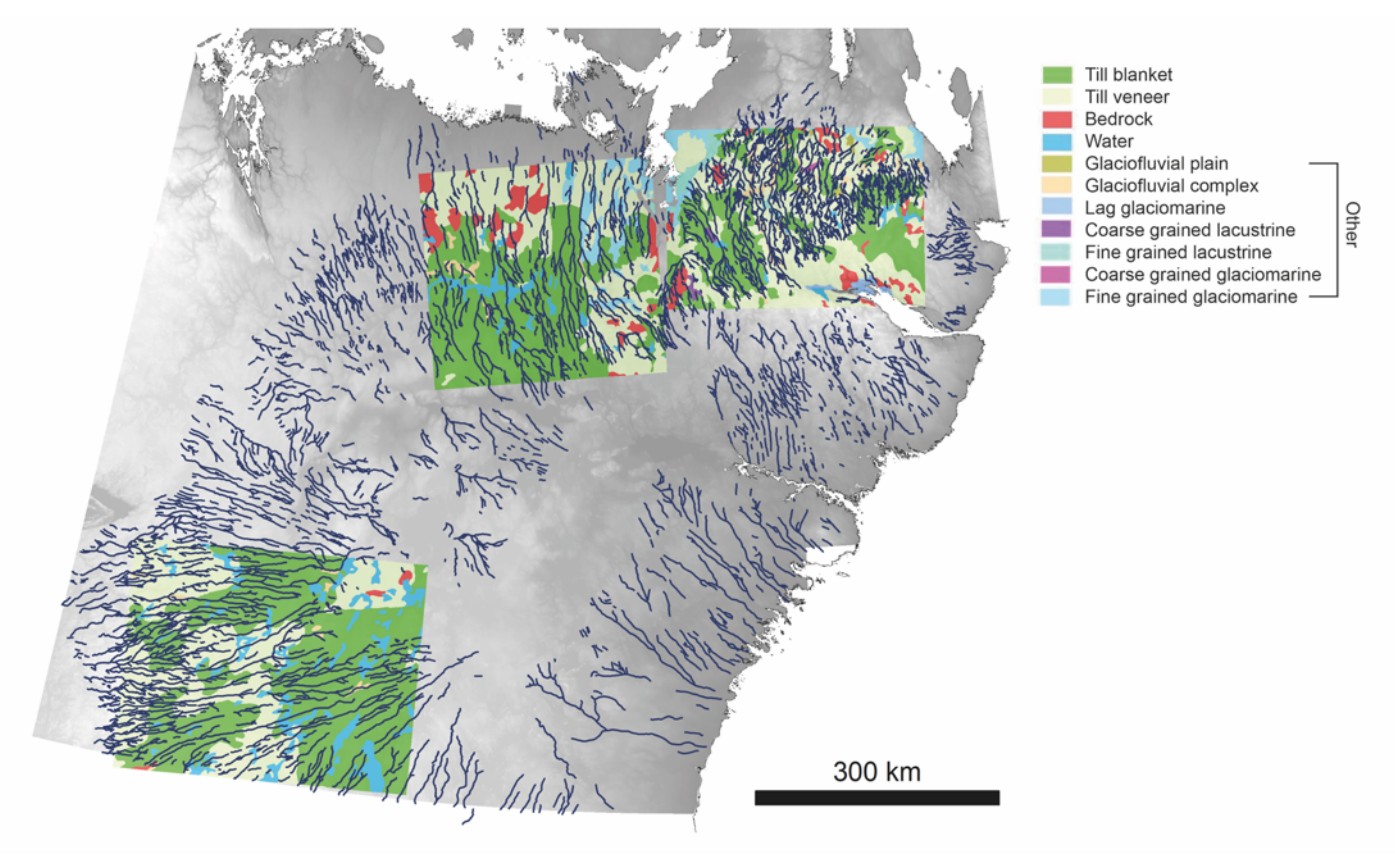

**Figure A2.** Surface substrate across the three test sites (left – right) used for analysis in
section 3.3

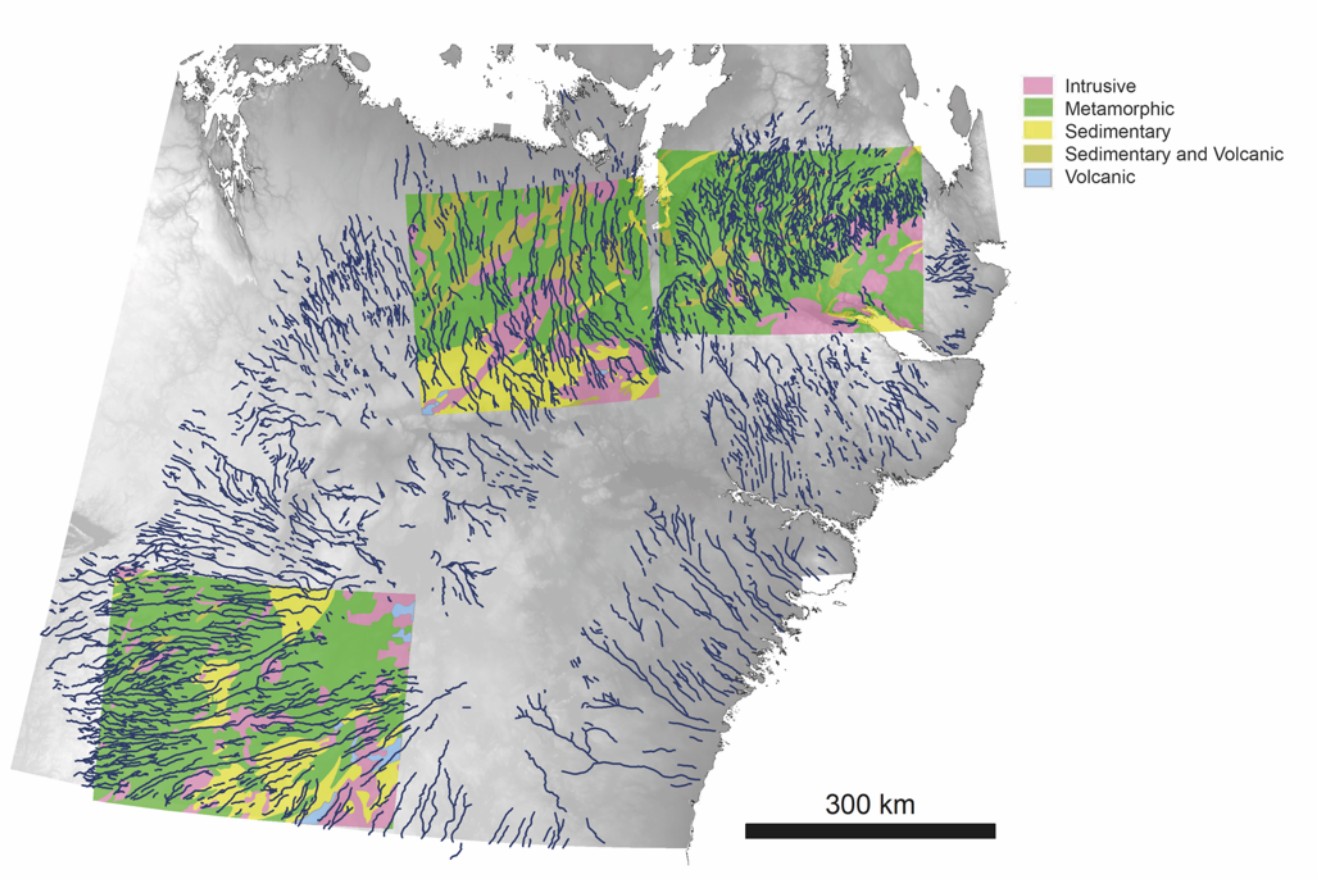

**Figure A3.** Bed geology across the three test sites (left to right) used for analysis in section 3.3