# Peer review of "A model for interaction between conduits and surrounding hydraulically connected distributed drainage based on geomorphological evidence from Keewatin, Canada."

_The Cryosphere, 2020_

## Referee Comment (RC1) · Anonymous Referee #1 · 27 Feb 2020

New imagery and DEMS made available through the Polar Geospatial Data Center have made it possible to map glacial landforms in a semi-automated approach and at a higher resolution than ever before. Without field-checking, these maps remain interpretations of landform origin and assumptions of the materials that comprise them.This paper does, however, cite some of the primary Canadian work on the ground so some of the interpretations have ground-truthing.

The focus of the work was on the concentration of a variety of landforms of interpreted subglacial fluvial origin in corridors and then an explanation for this. Areas previously

mapped as ice streams were noted to have fewer meltwater landforms. The suggestion that this was a matter of preservation potential of landforms in these locations was not addressed. The timing of the formation of the landforms can assumed by basic geologic principles. However, in this case, the authors suggested that the landforms represented approximately 1,000 years during deglaciation and were not overly concerned with a finer temporal resolution made possible by cross-cutting relationships or ice-marginal or ice-collapse features.

Without more geologic evidence, I was left with questions on: the contemporaneous nature and subglacial origin of features such as tunnels and similar-dimension positive-relief features; the subglacial origin and sedimentology of the hummocks; and the origin of splay landform which is similar to ice-marginal fans. I found the justification for their interpretations lacking.

However, this is part of the very nature of a paper based on the cataloging of landforms from remotely sensed data: questions will remain. Hypothesis for formation will need to be tested with observations of the materials at the very least. Measurements made on modern glaciers or demonstrated in experimental work would also help provide credibility to their interpretations that landforms vary with the pressurization of the subglacial water system. This is not a novel hypothesis, nor one that explains all observed features of the glacier bed but it is the primary interpretation of the paper.

The explanation for ice streams lacking meltwater features because of their low surface slopes is not the only plausible hypothesis; fast moving ice over deforming beds will destroy the evidence of channel formation. However, the authors emphasize that a well drained bed results in more stable ice. This is simply reversing the emphasis typically presented in ice-stream papers but using similar reasoning for the bimodal behavior of ice flow.

Non-standard and casual punctuation is pervasive throughout the paper: and/or, presence/absence, substrate/geology, splay/glaciofluvial, meltwater track/meltwater chan-

nel, erosion/deposition, streaming/surging. Please choose a conjunction or a single word. Hyphens are used in noun strings but not needed with adverbs. I would avoid unnecessary complicated acronyms: MW subscript route(s); MW subscript track(s), MW subscript corridors; VPA–variable pressure axis (a new one for me); GrIS for Greenland Ice Sheet (couldn't see where you defined that one). What is the point of using any of them in an online manuscript with no limits on characters or words? They are also not used consistently, e.g. line 516.

---

## Referee Comment (RC2) · Flavien Beaud (Referee) · 7 Mar 2020

In the proposed manuscript titled: Large-scale integrated subglacial drainage around the former Keewatin Ice Divide, Canada reveals interaction between distributed and channelized systems, by Lewington et al, the authors present a map of landforms associated with subglacial water flow in the Northwestern sector of the Laurentide Ice Sheet. In previous work (Lewington et al., 2019), the authors developed an automated method to extract such landforms from Digital Elevation Models (DEMs), method which they apply here to a sector of the Laurentide Ice Sheet. A motivation for this work is to analyze different landforms together and recognize their intertwined nature. This is a welcome advance in glacial geomorphology enabled by the availability of the Artcic-DEM, as studies of such landforms tend to overlook these interconnections. The authors perform a comprehensive mapping effort of the landforms and the relationships they find between different landforms will be essential in improving our understanding of their formation. I have a few points of concern, however, before this study can be published. The main concern is the proposed mechanism of formation of these landforms as a result of hydraulic changes in the vicinity of a main drainage channel. My second concern is that the presentation of the manuscript both in the text and the figures should be improved. In summary, I believe that the work presented is significant and, pending revisions, is publishable in The Cryosphere. Note that this general comment is associated with an annotated pdf of the manuscript where detailed comments can be found.

Scientific comments:

Generally, the literature cited is quite extensive, however, I think that the recent (past 5 years) literature on glacial processes has been a bit overlooked. For example, one the main point of the paper relies on the interactions between a subglacial channel and its surrounding drainage, topic that has been discussed a fair bit in recent papers (see Hoffman et al., 2016, Nature Communication; Stevens et al., 2015, Nature; Rada and Schoof, 2018, The Cryosphere; Andrews et al., 2014, Nature). While some of these papers are cited, their findings are not related in enough details. In terms of sediment transport, bedrock erosion and tunnel valleys and esker formations a few recent numerical modelling papers also offer some valuable insight as to how to interpret the landforms of interest here and their results should be discussed (see Beaud et al., 2016 Earth Surface Dynamics; Beaud et al., 2018, ESPL and JGR, Delaney et al., 2019, JGR). While a number of these paper are my research, I am not aware of any other study quantifying both sediment transport and bedrock incision by subglacial water flow. More details are listed in the commented pdf of the manuscript. The current manuscript does not provide enough evidence to substantiate the claim that the pressure fluctuations between a subglacial channel and its surroundings are responsible for the formation of subglacial landforms shaped by water. I think that there is, in fact, more evidence to show that the lateral migration of subglacial channels is a better candidate than the extent of the 'variable pressure axis'. The variable pressure axis idea is used to describe a temporary phenomenon, and a reversal in pressure gradient with respect to steady state conditions. As described in Hubbard et al. (1995), the water that flows outward from the channel during a pressure peak will subsequently flow back towards said channel. This mechanism cannot explain a net transport in either direction. A similar idea that does not involve the pressure gradient reversal is that channels tap into the surrounding distributed system within a distance that is linked to the hydraulic conductivity of the distributed drainage system (Hewitt, 2011, JoG; Hewitt, 2013, EPSL; Werder et al., 2013, JGR). This idea is used by Delaney et al. (2019, JGR) to explain the transport of material created by subglacial erosion toward a main channel, or by Hewitt and Creyts (2019) to determine the spacing of eskers. The size of the sediment transported in a distributed system is thought to be relatively small (Beaud et al., 2016, Earth Surface Dynamics), and may not be adequate to transport the sediment of the size found in the targeted landforms. It is common to find gravel if not boulders in the till. The flow forming these landforms should thus be able to transport such sediment. To explain the transport of significant amount of sediment or clasts of large size, floods resulting from lake drainages are often involved. However, for a single lake drainage to create a flood throughout a channel until it exists the glacier, the lake size has to be particularly large. For example, on the Greenland ice sheet lake drainages are unlikely to create large flood through a single channel as they are spread across a drainage catchment and may not drain at the same time (e.g. Bartholomew et al., 2012; Fitzpatrick et al., 2014). However, single drainage event will affect the bed in the vicinity (few kms) of the lake location (Stevens et al., 2015, Nature). Beaud et al. (2018, EPSL) further discusses the efficacy of floods of various magnitude to incise bedrock, and find that seasonal meltwater production is more likely to produce landforms like tunnel valleys.

Interactive
comment

To remove sediment or incise bedrock and form a negative-relief meltwater corridor, the water flow has to be able to produce shear stresses that are sufficient to transport coarse sand and gravel which are typical from glacial till. Therefore, the water flow velocities have to be sufficient and there should be enough accommodation space for these particles to be transported. Sediment transport in a macro-porous sheet has been modeled by Creyts &al. (2013, JGR) and sediment transport capacity in a network of connected cavities is discussed in Beaud et al. (2016 ESurf). In either study, the size of sediment that can be readily transported in a distributed system is small, i.e. on the order of size of coarse sand. I believe there is a strong body of evidence to suggest that the lateral migration of a subglacial channel across a melt water corridor is responsible for its formation. Gulley et al. (2012, JoG) find that the location of moulin has a large control on the location of the drainage channel and as a result of moulin advection, the said drainage channel changed paths from one year to the next. In the tunnel valley review presented in Kehew et al. (2012), there is ample evidence that the fill of tunnel valleys is constituted of numerous smaller individual channels, hinting a migration rather than a process occurring in a channel surroundings. Experiments of water flow under pressure over a sediment bed, show the braided nature of the resulting sediment channels (Catania and Paola, 2000, Geology) or the migration of single sediment channels (Lelandais et al., 2016, JGR). Finally, in this study, eskers are not necessarily at the center of a MWcorridor, whereas, in an isotropic medium and following the proposed VPA theory, they should. In an anisotropic medium, they should still be systematic characteristics on either side of a drainage channel likely dependent on the conductivity or the roughness of the substrate in this area. In summary, I believe that the VPA origin of landforms should be dropped, unless the authors are able to provide convincing evidence that it can produce sediment transport and bedrock erosion. However, I believe the author can make a strong case for subglacial channel migration being the responsible for the width of MWcorridors. Such migration has seldom been studied and the present study could thus open the door for the use of MWcorridor width as a proxy for glacial conditions. It is likely that the migration is limited by a combination of bed topography, ice surface topography, effective pressure (ice over-burden minus water pressure) and location of moulin. In addition, the spacing between MWcorridor is a more complete description of the esker or channel spacing proposed by Hewitt (2011) and Hewitt and Creyts (2019). Thus, the current results could actually be used to infer more robustly the spacing of drainage networks and thus inform the conductivity of the distributed drainage networks that should be used in numerical models.

Presentation:

Title change: the title should be modified to something that focuses more on the method and results than the interpretation of the results. Throughout the text there is copious use of comparatives, but they are not always compared to something specific. In general, please try to quantify these comparisons with numbers so that they are less subjective. It seems a bit weird to put eskers with splays and eskers at odds during the explanations. These are both positive relief features in opposition to the other negative relief landforms. Also, it seems circular to say that esker with splays are connected to eskers, since their name suggests they are eskers too. Please add some explanation of these choices in the manuscript. Many of the figure display text that is too small, or the maps show an area that is too large to see features clearly. Many figure panels also lack letters to reference them. Please add some to make the figure more intelligible. At the beginning of paper show examples of the different landforms and their manifestation on the DEM. I understand this is was the point of the Lewington et al., 2019 paper, but it is necessary to show this here again. Define splay around the esker in more details.

Best, Flavien Beaud

Please also note the supplement to this comment:
https://www.the-cryosphere-discuss.net/tc-2020-10/tc-2020-10-RC2-supplement.pdf

[Figure]

**Supplement:**

[revised manuscript text omitted]

- Erosional corridor (e.g. Burke et al., 2011)

- Esker corridor (e.g. Sharpe et al., 2017)

- Meltwater corridor (e.g. Rampton, 2000)

- Washed zone (e.g. Ward et al., 1997)

*'Positive':*

- Glaciofluvial corridor (e.g. St-Onge, 1984; Utting et al., 2009)

- Hummock corridor (positive) (e.g. Peterson and Johnson, 2018; Lewington et al., 2019)

- Esker with splay (e.g. Cummings et al., 2011) | *esker ridge* — erosion

*glaciofluvial deposits, anastomosing esker ridge, hummocks*

*hummocks*

*lateral esker ridge 'splay'* — deposition |
| Esker ridge | Single esker ridge | *esker ridge* |

[revised manuscript text omitted]

---

## Author Comment (AC2) · 20 Jun 2020

We thank the reviewer for their comments which have helped us to improve the manuscript. Their key issues were regarding our discussion of ice streams and with the relative timing and mechanism of landform formation. In response to this we have discussed the possibility of a lack of preservation of geomorphic signatures beneath ice streams in addition to the possibility that the dominant drainage mode and the resultant signature may be different here. In terms of our discussion on ice streams and the potential for a highly channelised bed to moderate ice velocities, previous geomorphic studies (e.g. Storrar et al., 2014) and examples in the contemporary (e.g. Sole et al., 2013; Williams et al. 2020) and modelling literature (e.g. Lelandais et al., 2018) support the concept that a well-drained bed (i.e. due to increased channelisation) has the ability to stabilise an ice sheet by regulating ice sheet velocity.

In response to the reviewer comments we have also improved the link between the proposed model and the geomorphic signature in the updated manuscript and have better grounded this within a growing body of theoretical, experimental and contemporary evidence, which supports; (a) the occurrence of over-pressurisation of subglacial conduits; and (b) the potential ability for geomorphic work as a result. The very nature of large-scale geomorphic studies such as this focus on providing information on where drainage does or does not occur and how differing signatures can be related to background controls. As such, we do not focus on finer resolution signatures and we believe that our updated manuscript now clearly explains the temporal aspect of formation within the model (i.e. corridors forming over 10's - 100's years within the ablation zone and eskers forming near the margin) and the assumptions involved when making comments about changes over time inferred from changes over space. In addition to this, we have reduced our use of acronyms throughout to make the paper easier to follow.

Reviewer comments are in black and our responses are in green.

Kind regards,

Emma Lewington (on behalf of all co-authors).

The focus of the work was on the concentration of a variety of landforms of interpreted subglacial fluvial origin in corridors and then an explanation for this. Areas previously mapped as ice streams were noted to have fewer meltwater landforms. The suggestion that this was a matter of preservation potential of landforms in these locations was not addressed.

Thank you for pointing this out. We have now included a sentence considering the possible lack of preservation of features formed in this area due to enhanced erosion of fast flowing ice (e.g. Boulton, 1996). We also acknowledge that the presence of soft floored sediments may not result in the same geomorphic signature (i.e. water may be stored in till or organised into braided canals), and thus the absence of meltwater routes does not explicitly indicate the absence of efficient flow.

The timing of the formation of the landforms can be assumed by basic geologic principles. However, in this case, the authors suggested that the landforms represented approximately 1,000 years during deglaciation and were not overly concerned with a finer temporal resolution made possible by cross-cutting relationships or ice-marginal or ice-collapse features.

Within the study area we find no evidence of cross cutting of meltwater corridors or eskers. We do find eskers within meltwater corridors and have added in a section to make the temporal aspect of our model clearer - i.e. meltwater corridors were eroded within the ablation zone due to repeated pressure fluctuations over 10's to 100's years while eskers were most likely deposited near the ice margin. This paper presents a large-scale study and we are interested in the overall trend. We agree that more detailed morphology (e.g. esker splays and anastomosing esker sections) could be informative for high resolution ice margin retreat (e.g. Livingstone et al., 2020) but this is beyond the scope of this paper.

Without more geologic evidence, I was left with questions on: the contemporaneous nature and subglacial origin of features such as tunnels and similar-dimension positive- relief features; the subglacial origin and sedimentology of the hummocks; and the origin of splay landform which is similar to ice-marginal fans. I found the justification for their interpretations lacking. However, this is part of the very nature of a paper based on the cataloguing of land- forms from remotely sensed data: questions will

remain.

A subglacial meltwater origin has been established for meltwater channels and tracks within the literature (e.g. Rampton, 2000; Utting et al., 2009; Peterson and Johnson, 2018) and we are not the first to suggest a potential continuum between these features (e.g. Sjogren et al., 2002; Peterson and Johnson, 2018). However, we agree that we may have overstated the importance of eskers surrounded by lateral splays as a potential positive relief expression of this. Furthermore, we acknowledge that there are issues grouping eskers with lateral splays with the meltwater channels and tracks as some or all of these features may have formed subaerially or even supraglacially at the ice sheet margin rather than subglacially. Therefore, we no longer include eskers with lateral splays within the term meltwater corridor (Table 2).

Hypothesis for formation will need to be tested with observations of the materials at the very least. Measurements made on modern glaciers or demonstrated in experimental work would also help provide credibility to their interpretations that landforms vary with the pressurization of the subglacial water system.

We agree that our proposed model for formation of meltwater corridors will need to be tested and we hope that publication of our paper might prompt such work. In terms of modifications to the paper, we have updated the proposed model section to demonstrate the increasing evidence from contemporary and modelling studies, which support the occurrence of conduit over-pressurisation and the potential for this to access and erode sediment from the bed. This process has been inferred from simultaneous field observations of ice velocity and uplift, surface melt and proglacial river discharge in contemporary settings and we suggest is likely to have occurred in Keewatin, an area located on resistant shield bedrock that would have experienced high surface meltwater production rates during the time period covered by our study. Therefore, we think there is sufficient evidence to support the plausibility of our model as a simple explanation of a wide range of geomorphic signatures.

This is not a novel hypothesis, nor one that explains all observed features of the glacier bed but it is the primary interpretation of the paper.

We disagree with the first part of this statement and believe we now provide sufficient evidence from the geomorphic and contemporary ice sheet and alpine glacier

literature to support the conceptual model proposed within this study. We believe this paper and our proposed model provides support for viewing geomorphic evidence of subglacial meltwater landforms as a holistic signature and provides a more realistic understanding of the distribution of drainage beneath ice sheets.

We agree that our model does not 'explain all observed features of the glacier bed'; it would be a large task indeed to explain everything and we don't claim to.

The explanation for ice streams lacking meltwater features because of their low surface slopes is not the only plausible hypothesis; fast moving ice over deforming beds will destroy the evidence of channel formation.

We agree and this has now been modified (please see first comment response).

However, the authors emphasize that a well-drained bed results in more stable ice. This is simply reversing the emphasis typically presented in ice-stream papers but using similar reasoning for the bimodal behaviour of ice flow.

There is evidence from the contemporary literature, which suggests that gradual dewatering of the bed is occurring in response to increased surface melt and inferred subglacial channelisation, which results in an observed year-on-year reduction in ice sheet velocity (e.g. Sole et al., 2013; Tedstone et al., 2015) and vice versa (Williams et al. 2020). Although this a temporal relationship (i.e. when surface melt is higher and subglacial drainage more channelised, regional ice velocity is lower) and our argument for Keewatin is spatial (i.e. where there is evidence for more channelised drainage there are fewer ice streams (e.g. Storrar et al., 2014)), we are invoking the same mechanism. This idea is also consistent with modelling of Lelandais et al., (2018), who suggest that efficient drainage (i.e. the development of tunnel valleys) can shut down ice streams and may be crucial for stabilising ice sheets during periods of climate change.

Non-standard and casual punctuation is pervasive throughout the paper: and/or, presence/absence, substrate/geology, splay/glaciofluvial, meltwater track/meltwater channel, erosion/deposition, streaming/surging. Please choose a conjunction or a single word. Hyphens are used in noun strings but not needed with adverbs. I would avoid unnecessary complicated acronyms: MW subscript route(s); MW subscript

track(s), MW subscript corridors; VPA–variable pressure axis (a new one for me); GrIS for Greenland Ice Sheet (couldn't see where you defined that one). What is the point of using any of them in an online manuscript with no limits on characters or words? They are also not used consistently, e.g. line 516.

We understand that the use of acronyms may have been frustrating and made the paper more difficult to follow. We have now reduced the number of acronyms used and have made sure to state clearly what they refer to when using them for the first time. We have also amended the non-standard and casual punctuation throughout.

**References:**

Boulton, G.S. The origin of till sequences by subglacial sediment deformation beneath mid-lattitude ice sheets. Annals of Glaciology. 22: 75-84.

Lelandais, T. Ravier, E. Pochat, S. Bourgeois, O. Clark, C. Mourgues, R. Strzerzynski, P. Modelled subglacial floods and tunnel valleys control the life cycle of transitory ice streams. The Cryosphere. 12: 2759-72. 2018.

Livingstone, S.J. Lewington, E.L.M. Clark, C.D. Storrar, R.D. Sole, A.J. McMartin, I. Dewald, N. Ng, F.. A quasi-annual record of time-transgressive esker formation: implications for ice sheet reconstruction and subglacial hydrology. The Cryosphere. 2020.

Peterson, G. Johnson, M.D. Hummock corridors in the south-central sector of the Fennoscandian ice sheet, morphometry and pattern. Earth Surface Processes and Landforms. 43: 919-29. 2018.

Rampton, V.N. Large-scale effects of subglacial meltwater flow in the southern Slave Province, Northwest Territories, Canada. Canadian Journal of Earth Sciences. 37(1). 81–93. 2000.

Sjogren, D.B. Fisher, T.G. Taylor, L.D. Jol, H.M. Munro-Stasiuk, M.J. Incipient tunnel channels. Quaternary International. 90. 41–56. 2002.

Sole, A. Nienow, P. Bartholomew, I. Mair, D. Cowton, T. Tedstone, A. et al., Winter motion mediates dynamic response of the Greenland Ice Sheet to warmer summers. Geophysical Research Letters. 40(15). 3940–44. 2013.

Storrar, R.D., Stokes, C.R. Evans, D.J.A. Increased channelization of subglacial drainage during deglaciation of the Laurentide Ice Sheet. Geology. 42(3). 239–42. 2014.

Tedstone, A.J. Nienow, P.W. Gourmelen, N. Dehecq, A. Goldberg, D. Hanna, E. Decadal slow-down of a land-terminating sector of the Greenland Ice Sheet despite warming. Nature. 526: 692 – 95. 2015.

Utting, D.J., Ward, B.C. Little, E.C. Genesis of hummocks in glaciofluvial corridors near the Keewatin Ice Divide, Canada. Boreas. 38(3). 471–481. 2009.

Williams, J.J. Gourmelen, N. Nienow, P. Dynamic response of the Greenland ice sheet to recent cooling. Scientific Reports. 10: 1647.

---

## Author Comment (AC3) · 20 Jun 2020

We thank Flavian Beaud for his very thoughtful comments; in response, we have made significant revisions, which we think have helped us to clarify and refine the model, and to improve the paper. The key suggestion was that the VPA model should be dropped and the lateral migration theory expanded upon. This is based largely on scepticism as to whether geomorphic work can occur across the VPA (i.e. whether sediment can be eroded and removed). In response, we have made a number of significant changes to bolster our argument and to make the case that both ideas are compatible, including:

(i)     Expanding on the modern glacial literature that provides evidence that conduit over-pressurisation occurs on daily to seasonal timescales, and that this can result in the conduit being overwhelmed and drainage reconfigured, for example as a sheet flood (Russell et al., 2007) or as anastomosing or braided channels/canals (e.g. Catania & Paola, 2001; Gulley et al., 2012; Vore et al., 2019). The idea of overpressurisation resulting in braiding or anastomosing is also consistent with the idea that conduits can transiently migrate laterally and we therefore think that we are roughly on the same page but that this was not well-explained in the original submission.

(ii)    Engaging with the glacial erosion literature, which provides strong support that over-pressurisation events can access and mobilise sediments beyond the subglacial conduit (e.g. see Delaney et al., (2018) who propose a similar mechanism to explain variability in subglacial sediment transport for two glaciers in the Swiss Alps), and that eskers scavenge sediments locally (e.g. Bolduc, 1992).

(iii)    Better defining the timescales at which the observed features formed (10s to 100s of years), and the implication that they are therefore a composite signature produced by repeated pressure perturbations.

(iv)    Clearly defining and consistently using the more descriptive term '*hydraulically connected distributed drainage system*' (we consider it to be the lateral limit of the influence of pressure variations that originate in a subglacial conduit that can result in water flow into or out of the conduit) instead of 'VPA' to provide greater clarity about what process we are referring too (and this clearly differentiates it from the weakly connected distributed drainage system where we don't expect geomorphic work to

occur). To further help here we also discuss the different uses of the term in the literature in the introduction (e.g. 'VPA' in Hubbard et al., 1995; 'hydrological subcatchments' in Rada and Schoof, 2018; 'efficient core' in Davison et al., 2019).

In addition to these arguments, we have further expanded the discussion on lateral channel migration, which we agree could occur, particularly where the hydraulic gradient is lower, higher up the ice sheet (e.g. see Vore et al., 2019). Indeed, this idea is also consistent with our model that during overpressurisations drainage can transiently reconfigure (e.g. by migrating laterally) (point i, above). We do, however, suggest that lateral migration of the principle channel is less likely to occur nearer the margin, based on modern observations of conduit stability (Vore et al., 2019) and the simple observation that eskers (relict conduits) and the surrounding meltwater corridors are straight and line up along-flow (e.g. see Storrar et al., 2014); if the conduit migrated extensively near the margin you would expect to find extremely sinuous eskers or esker sections that are offset. We therefore maintain that the more general VPA idea is the simplest explanation of the landform relationship we observe (although we also highlight the potential for conduit migration, which could occur as part of drainage reconfiguration, or instead of the VPA higher up the ice sheet), and fits with modern observations of frequent conduit overpressurisation (e.g. Bartholomew et al., 2011; Rada and Schoof, 2018), scales with modelling and observations of the width-range of the VPA in Greenland (e.g. Werder et al., 2013; Tedstone et al., 2015) and agrees with and helps explain sediment transport observations (e.g. Delaney et al., 2018).

By publishing this model, we hope to stimulate future modelling studies and palaeo and contemporary observations to test both our preferred model, and alternatives, which we do not rule out. We expand on these points below in response to the specific comments made by the reviewer.

Reviewer comments are in black and our responses are in green.

Kind regards,

Emma Lewington (on behalf of all co-authors)

A motivation for this work is to analyze different landforms together and recognize their intertwined nature. This is a welcome advance in glacial geomorphology enabled by the availability of the ArtcicDEM, as studies of such landforms tend to overlook these interconnections. The authors perform a comprehensive mapping effort of the landforms and the relationships they find between different landforms will be essential in improving our understanding of their formation.

Thank you for these comments.

Generally, the literature cited is quite extensive, however, I think that the recent (past 5 years) literature on glacial processes has been a bit overlooked. For example, one the main point of the paper relies on the interactions between a subglacial channel and its surrounding drainage, topic that has been discussed a fair bit in recent papers (see Hoffman et al., 2016, Nature Communication; Stevens et al., 2015, Nature; Rada and Schoof, 2018, The Cryosphere; Andrews et al., 2014, Nature). While some of these papers are cited, their findings are not related in enough details.

Thank you for pointing this out, we have now expanded the section in the introduction discussing the three-system drainage model as the accepted current state of understanding and have also expanded on the interactions occurring over a range of timescales in more detail. To help this we have produced a new figure to highlight the key terms.

In terms of sediment transport, bedrock erosion and tunnel valleys and esker formations a few recent numerical modelling papers also offer some valuable insight as to how to interpret the landforms of interest here and their results should be discussed (see Beaud et al., 2016 Earth Surface Dynamics; Beaud et al., 2018, ESPL and JGR, Delaney et al., 2019, JGR). While a number of these papers are my research, I am not aware of any other study quantifying both sediment transport and bedrock incision by subglacial water flow.

Thank you for these recommendations, we have now referred to these papers with regards to the importance of seasonal water flows in the development of tunnel valleys.

The current manuscript does not provide enough evidence to substantiate the claim

that the pressure fluctuations between a subglacial channel and its surroundings are responsible for the formation of subglacial landforms shaped by water.

We agree that we may not have been sufficiently clear about how our proposed model suggests water pressure fluctuations within the hydraulically connected distributed drainage system contribute to landform formation. It is during periods when the conduit connects to the hydraulically connected distributed drainage system that we expect most geomorphic work to occur. Indeed, observations of subglacial fluvial sediment transport (Swift et al., 2002) and lithological investigations of eskers (e.g. Bolduc, 1992) suggest that subglacial streams access sediment beyond the small fraction of the bed they immediately occupy. In particular, and following similar observations at other glaciers (e.g. Collins, 1990, 1996; Swift et al., 2002), Delaney et al. (2018) observed increases in sediment discharge at two alpine glaciers during times when water discharge increased rapidly (lake drainages and precipitation events). They readily explain this by invoking the same mechanism proposed in our study (i.e., rapid water inputs overwhelm the drainage system, creating high water pressures and causing both increased access of water to sediment sources and high enough water velocity to transport sediment). We therefore stick by our original supposition that sudden variations in basal water pressure do result in geomorphic work, noting that even precipitation events are able to cause an increase in sediment mobilisation (Delaney et al., 2018), and that the corridors are a composite imprint formed over 10s - 100s of years. We have now expanded the discussion to include a section on subglacial fluvial erosion, where we discuss how our model fits with observations and modelling from this wider literature.

I think that there is, in fact, more evidence to show that the lateral migration of subglacial channels is a better candidate than the extent of the 'variable pressure axis'. The variable pressure axis idea is used to describe a temporary phenomenon, and a reversal in pressure gradient with respect to steady state conditions. As described in Hubbard et al. (1995), the water that flows outward from the channel during a pressure peak will subsequently flow back towards said channel. This mechanism cannot explain a net transport in either direction. A similar idea that does not involve the pressure gradient reversal is that channels tap into the surrounding distributed system within a distance that is linked to the hydraulic conductivity of the distributed drainage

system (Hewitt, 2011, JoG; Hewitt, 2013, EPSL; Werder et al., 2013, JGR). This idea is used by Delaney et al. (2019, JGR) to explain the transport of material created by subglacial erosion toward a main channel, or by Hewitt and Creyts (2019) to determine the spacing of eskers.

We disagree with these points. Firstly, a pressure reversal can cause net erosion if the energy (i.e. ability to erode, transport etc.) of the water as it flows away from the channel is different to when it flows back towards the channel. For example, if a surface lake drains, the increase in water pressure in a connected subglacial channel might occur over ~1 hour, and depending on the downstream hydropotential gradient, the water pressure may then gradually decrease over tens of hours (e.g. inferred from uplift plots in Fig. 3 of Doyle et al. 2013). The ability of water flow induced by the lake drainage to erode and transport sediment will be much greater than that of the returning water flow. Water also does not just flow laterally - when the conduit is overwhelmed water will continue to drain towards the margin, but over a broader area (e.g. Catania and Paola, 2001; Russell et al., 2007; Gulley et al., 2012; and see comment below), analogous to when a river bursts its banks during a flood event. So, sediment eroded at one point in the hydraulically connected distributed drainage system will be transported down the regional hydraulic gradient towards the ice margin.

We consider the VPA (now termed the hydraulically connected distributed drainage system) to be the lateral limit of the influence of pressure variations that originate in a subglacial channel. This lateral influence of the conduit on its surroundings therefore encompasses and supports the modelling ideas of Hewitt, Werder, Delaney and others, and also of a wide body of glacier and ice sheet observations demonstrating a hydraulic connection between the conduit and its surroundings (e.g. Rada and Schoof, 2018; Nanni et al., 2020).  We have attempted to address these points in an expanded introduction that includes a new figure with key drainage terms defined, and a more complete description of this connection, including how the lateral limit varies depending on the magnitude of the pressure perturbation (determined by the volume and rate of meltwater input), antecedent conditions (e.g. the limit will decline on average through a melt season as the conduits become more efficient - see Rada and Schoof, 2018) and basal substrate.

We agree conduit migration is certainly possible, and indeed, also consistent with the idea of transient drainage reconfiguration during overpressurisation (e.g. Catania & Paola, 2001; Gulley et al., 2012); we have expanded this section to discuss these points further and to better link to our overall model. In particular, there is some recent evidence that where the hydraulic potential is low (e.g. higher up the glacier) drainage occurs through multiple conduits, which migrate over time (see Vore et al., 2019). However, nearer the ice margin, where the hydraulic potential is steeper, the same research suggests the conduit is relatively stable in space, and this fits with the simple geomorphic observation that eskers (relict conduits) and the surrounding meltwater corridors are straight and line up (e.g. see Storrar et al., 2014 who found a median sinuosity of 1.04 for a sample of >20,000 Canadian eskers). To do this the conduit needs to be centred in a stable location. If the conduit migrated extensively you would expect to find extremely sinuous eskers or eskers sections that are offset. In addition, the meltwater corridors themselves are relatively straight with a narrow width range.

The size of the sediment transported in a distributed system is thought to be relatively small (Beaud et al., 2016, Earth Surface Dynamics), and may not be adequate to transport the sediment of the size found in the targeted landforms. It is common to find gravel if not boulders in the till. The flow forming these landforms should thus be able to transport such sediment.

We agree that in a passive or weakly connected distributed drainage system sediment transport is generally likely to be low and dominated by or exclusively gravel or smaller. However, our model is focused on the impact of conduit overpressurisation on the hydraulically connected distributed drainage system. When the conduit becomes overwhelmed during rapid water pressure perturbations, the conduit is shown to 'connect' to this wider area (e.g. Hubbard et al., 1995; Bartholomew et al., 2001; Rada and Schoof, 2018). When it does so, flow is no longer thought to occur as 'passive' distributed drainage, but somewhere along a continuum between classical distributed and channelised drainage depending on the pressure gradient between the conduit and its surroundings. The amount of geomorphic work at that time likely depends on where on this continuum the drainage event was (and of course we see a composite of many such events). Observations and modelling suggest this may range from cavity

expansion, to conduit reconfiguration, to local sheet flood (e.g. Cowton et al., 2016; Catania and Paola, 2001; Russell et al., 2007; Gulley et al., 2012). To make this distinction clearer we have expanded the introduction section to more clearly define the three-system drainage model (i.e. conduit, hydraulically connected and weakly connected drainage systems), and produced a new model figure that details the evolution of the VPA/ well connected drainage system to pressure perturbations (in Fig. 14 b1-b3). We have also carefully defined all our terms and used these consistently throughout. Please also see our above comment in regards to the ability of these overpressurisation events to cause geomorphic work.

To explain the transport of significant amounts of sediment or clasts of large size, floods resulting from lake drainages are often involved. However, for a single lake drainage to create a flood throughout a channel until it exits the glacier, the lake size has to be particularly large. For example, on the Greenland ice sheet lake drainages are unlikely to create large floods through a single channel as they are spread across a drainage catchment and may not drain at the same time (e.g. Bartholomew et al., 2012; Fitzpatrick et al., 2014). However, single drainage events will affect the bed in the vicinity (few kms) of the lake location (Stevens et al., 2015, Nature).

A supraglacial lake drainage may do significant geomorphic work locally, but once the moulin is opened up, suddenly there is a direct connection to surface-generated water, the amount of which varies substantially on diurnal etc. timescales. We know that this variability in discharge (increases of up to 50 cumecs) does reach the margin because we see it in proglacial discharge records (e.g. Bartholomew et al., 2011). This variability clearly overpressurises the subglacial channel (and thus could lead to geomorphic work) because it leads to hydrologically-driven diurnal ice velocity variations (e.g. Bartholomew et al., 2012). In addition, the rapid increase in discharge is sufficient to move cobbles and sometimes boulders as bedload in the proglacial river. Although this may be more difficult subglacially due to accommodation space, evidence of locally transient uplift of up to 1 m (e.g. Doyle et al., 2013) suggest that significant bedload could be moved (see also reply to comment below).

Meltwater is transported to the ice bed by hundreds of moulins and crevasses (unless it is stored in a lake) and so along the length of a subglacial channel, there will be

diurnal water pressure variations caused by surface-derived meltwater entering that channel from the moulins and crevasses. Thus, the combined effect of all this water reaching the bed is for there to be pressure variations sufficient to (probably) do geomorphic work along the whole length (or the vast majority of the length) of a subglacial channel.

Beaud et al. (2018, EPSL) further discusses the efficacy of floods of various magnitude to incise bedrock, and find that seasonal meltwater production is more likely to produce landforms like tunnel valleys. To remove sediment or incise bedrock and form a negative-relief meltwater corridor, the water flow has to be able to produce shear stresses that are sufficient to transport coarse sand and gravel which are typical from glacial till. Therefore, the water flow velocities have to be sufficient and there should be enough accommodation space for these particles to be transported. Sediment transport in a macro-porous sheet has been modelled by Creyts et al. (2013, JGR) and sediment transport capacity in a network of connected cavities is discussed in Beaud et al. (2016 ESurf). In either study, the size of sediment that can be readily transported in a distributed system is small, i.e. on the order of size of coarse sand. I believe there is a strong body of evidence to suggest that the lateral migration of a subglacial channel across a melt water corridor is responsible for its formation. Gulley et al. (2012, JoG) find that the location of moulin has a large control on the location of the drainage channel and as a result of moulin advection, the said drainage channel changed paths from one year to the next. In the tunnel valley review presented in Kehew et al. (2012), there is ample evidence that the fill of tunnel valleys is constituted of numerous smaller individual channels, hinting a migration rather than a process occurring in a channel surroundings. Experiments of water flow under pressure over a sediment bed, show the braided nature of the resulting sediment channels (Catania and Paola, 2000, Geology) or the migration of single sediment channels (Lelandais et al., 2016, JGR).

We acknowledge that we may have induced some confusion due to a lack of consistency in our terms used in the original paper. However, we are not suggesting that the high amounts of sediment erosion or transport occur within the weakly connected distributed drainage system (i.e. the passive system) but instead within the hydraulically connected distributed system during periods of overpressurisation when

water is forced out of the conduit and into the surrounding distributed system, resulting in drainage reorganisation in the form of anything from cavity expansion to a local sheet flood. In this scenario, consequent local hydraulic jacking of the ice would generate accommodation space (up to 1 m locally – Doyle et al. 2013) and sufficient water velocities to erode and entrain sediment (i.e. forming the meltwater corridors probably through repeated drainages over 10s to 100s years). In the weakly-connected distributed drainage system (i.e. areas in between the meltwater corridors with passive distributed drainage) we agree that we do not expect to see any visible evidence of fluvial erosion, and indeed we see this from the glacial geomorphological imprint.

Please see above our detailed comments about the role of conduit migration in the formation of these features. While we do not discount this as a possibility, we suggest that the processes within the VPA model are supported by more evidence from contemporary ice sheets and fit with the geomorphological observations. We do note that the work of Gulley et al. (2012) and Catania and Paola (2000) references here seem to support our model that during periods of overpressurisation drainage can reconfigure to form anastomosing or braided conduit networks (see comments above). Although it is true that the location of moulins are thought to be a key control on drainage pathway, there is strong evidence that moulins typically reset to approximately the same location as streams are captured upstream of the existing moulin by new crevasses which occur in the same position as the original moulin formed (due to constant stresses in the ice over the timescale of moulin formation) (see e.g. Catania and Neumann, 2010; and the Gulley paper also makes this point).

Finally, in this study, eskers are not necessarily at the center of a MWcorridor, whereas, in an isotropic medium and following the proposed VPA theory, they should. In an anisotropic medium, they should still be systematic characteristics on either side of a drainage channel likely dependent on the conductivity or the roughness of the substrate in this area.

We would suggest that the fact that we are dealing with reality and not a model (isotropic medium) contributes additional complexities, so that we would not expect the eskers to always (or ever) sit in the centre of a corridor. We would expect the lateral extent of the VPA to vary depending on the regional and local hydraulic potential

surface. If the pre-existing hydraulic gradient is steep and positive away from one side of the channel, the VPA on this side would be narrow. If the pre-existing hydraulic gradient on the other side is positive but shallow, the VPA on this side would be relatively wide. To reuse our above analogy; a subaerial river whose channel lies closer to one valley side will have an uneven flood extent on either side of the channel.

In summary, I believe that the VPA origin of landforms should be dropped, unless the authors are able to provide convincing evidence that it can produce sediment transport and bedrock erosion. However, I believe the author can make a strong case for subglacial channel migration being the responsible for the width of MWcorridors. Such migration has seldom been studied and the present study could thus open the door for the use of MWcorridor width as a proxy for glacial conditions. It is likely that the migration is limited by a combination of bed topography, ice surface topography, effective pressure (ice over- burden minus water pressure) and location of moulin.

We have not dropped our case for the VPA origin of landforms, but have instead clarified our phrasing and added additional evidence to support the proposed model (see above comments for more detail). We now think that the VPA model is clearer and more convincing and is supported by a range of observations (of over-pressurisation and sediment transport) from contemporary ice sheet and alpine settings. We acknowledge that conduit migration probably also plays a role in the formation of these features (particularly where the hydraulic gradient is shallower) and actually think this is consistent with our model and one possible outcome of conduit overpressurisation as part of a spectrum of possible responses (covering cavity expansion, braiding, anastomosing and sheet floods).

In addition, the spacing between MWcorridor is a more complete description of the esker or channel spacing proposed by Hewitt (2011) and Hewitt and Creyts (2019). Thus, the current results could actually be used to infer more robustly the spacing of drainage networks and thus inform the conductivity of the distributed drainage networks that should be used in numerical models.

We agree and think that this is an exciting key output of this paper. Our results suggest that surface meltwater inputs influence between 5 - 36 % of the bed (meltwater corridor coverage) which is 25 times more than conduits (eskers) alone (~ 0.5 % in this area).

This has implications for understanding the impact of surface meltwater inputs on ice sheet velocity and for sediment access and removal at the bed. The fact that not all meltwater corridors contain eskers supports the idea that eskers are a minimum map and that including additional lines of evidence (i.e. a holistic drainage map) is likely to generate a more realistic impression of the true drainage coverage. Finally, we have added a new figure which shows that the centreline mapping results from this study are within the range of other studies (many of which are in the Keewatin area), which have relied on eskers alone, but that our mean (8.1 km) is towards the lower end of this range. Interestingly, the average spacing appears to decrease with studies over time, perhaps indicating that increased resolution datasets (and in this case detailed mapping of all visible subglacial meltwater features) and large scale mapping studies are reducing the perceived spacing between meltwater routes.

Title change: the title should be modified to something that focuses more on the method and results than the interpretation of the results.

We agree that the title does not currently reflect the paper as well as it could and have thus changed it to 'A model for interaction between conduits and surrounding hydraulically connected distributed drainage system based on geomorphological evidence from Keewatin, Canada.' Although this still focuses on the interpretation of the results, we now make clear in the title that this is a model, which we hope will stimulate future modelling studies and palaeo and contemporary observations to test.

Throughout the text there is copious use of comparatives, but they are not always compared to something specific. In general, please try to quantify these comparisons with numbers so that they are less subjective. It seems a bit weird to put eskers with splays and eskers at odds during the explanations. These are both positive relief features in opposition to the other negative relief landforms. Also, it seems circular to say that esker with splays are connected to eskers, since their name suggests they are eskers too. Please add some explanation of these choices in the manuscript.

We agree that we may have overstated the importance of eskers surrounded by lateral splays as a potential positive relief expression of the negative relief landforms and have reduced the discussion on this (please see more detailed comments in response to reviewer one on this). We have also made an effort to be less subjective in our

comparisons.

Many of the figures display text that is too small, or the maps show an area that is too large to see features clearly. Many figure panels also lack letters to reference them. Please add some to make the figure more intelligible.

We have amended these concerns.

At the beginning of the paper show examples of the different landforms and their manifestation on the DEM. I understand this is was the point of the Lewington et al., 2019 paper, but it is necessary to show this here again. Define splay around the esker in more details.

We have now added a new figure with zoomed in examples of each of the geomorphic expressions of subglacial meltwater flow that we discuss in the introduction.

**Specific comments:**

Thank you very much for the detailed comments on the manuscript these have been addressed as follows:

14. Please define acronym when first used

We have now done this.

15. "Eskers exhibit key similarities with [...] association with eskers"? Types of what? That sentence should be reworded to be more clear.

We have now done this.

23. The sentence construction in the abstract is a bit confusing and sometimes convoluted. Please rewrite to make it more direct.

We have re-written much of the abstract to make it clearer.

45. I think the wording is not quite right. The efficient drainage can have a relatively large effect on ice motion, but that effect is to reduce sliding rather than enhance it.

We have re-written this sentence as follows: 'The configuration of the subglacial hydrological system is key to this, with a hydraulically efficient drainage system able to accommodate and evacuate an equivalent water flux without causing spikes in basal water pressure which have been linked to transient ice accelerations (e.g. Tedstone et al., 2013).'

53. I think it is worth including more recent literature like Hoffman et al. (2016, Nature), as it tackles that idea that the system is not binary. Hoffman is referred to in the next paragraph, I believe it should be done earlier though. The paper by Rada & Schoof, should also be discussed here. I general the literature cited in this paragraph should described a bit more extensively to reflect their findings. The main point is that the connectivity of the distributed drainage has a key effect on overall water pressures and sliding speed. I believe this is relevant for this study.

We have expanded this section significantly in the introduction and have included the key references you have recommended. We now make it clear that the subglacial drainage system is comprised of three main components: (i) a moulin connected channelised system; (ii) an active hydraulically connected distributed system influenced by the channelised system and; (iii) a weakly connected distributed system largely isolated and rarely influenced by surface meltwater (e.g. Andrews et al., 2014; Hoffman et al., 2016; Rada and Schoof, 2018).

We have also added in a new schematic to conceptualise this.

83. Beaud et al. (2018, ESPL) and Beaud et al. (2018, JGR) use numerical models to explore the formation tunnel valleys and eskers respectively. While it may be weird to point to my own work, these two papers are the only studies I am aware of that explores the processes responsible sediment transport and erosion by water flow under glaciers in details. The findings are directly relevant to the interpretations made in this study.

We agree and have referenced both papers in this section.

112. over-pressurization with respect to what? over atmospheric pressure or over ice-overburden pressure?

We have rephrased to 'during periods of high water pressure within the conduit'

123. Please also look at Catania & Paola (2000) and Lelandais et al 2016, where tunnel valley development is explored with physical experiments.

Thank you for these recommendations we have now included both in our introduction to tunnel valleys.

129. Beaud et al. 2016/ 2018 propose, 3) that floods are unnecessary, because subglacial water pressure gradients are large enough with seasonal meltwater flow to produce shear stresses that can explain sed transport and incision.

We have added the following to address this: 'iii) formation from seasonal meltwater flow (Beaud et al., 2016, 2018b).'

135. I think the distinction between features typical from bedrock or sediment substrates should be more clear or at least discussed. The processes of subglacial water flow expected are vastly different as a function of said substrate.

We have modified the section discussing tunnel valleys to acknowledge this: 'Tunnel valleys typically occur on soft substrate and are observed to occur at various developmental stages from mature and clearly defines to indistinct valleys often associated with hummocky terrain or as a series of aligned (e.g. Kehew et al., 1999; Sjogren et al., 2002). Bedrock carved tunnel valleys are rarer and tend to be narrower and more V-shaped in cross-profile (Van der Vegt et al., 2012).'

137. Sediment and ice-walled conduits have fundamentally different characteristics. If blended under the same term, this should be explained in details.

We are clarifying terminology not process understanding here and thus have decided not to expand into a discussion here.

163. I think it is worth recognizing that, in particular for tunnel valleys many of them might be buried under sediment, i.e. their current topographical expression is not necessarily representative of their actual presence.

We have added in the following sentence to address this: 'Importantly, we note this is a minimum map as some landforms – particularly tunnel valleys – may be fully or partially buried (e.g. Jørgensen and Sandersen, 2006).'

172. Please give some quantification of 'low' here.

We have rephrased this sentence to say 'local negligible relief'. Shilts et al., (1987) suggest that the relief rarely exceeds 95 m and is generally subdued.

173. Resistant to what? The quantification of the 'resistance' of bedrock can also be difficult if thought of in term of erosion. A number of studies show that fracture density rather than lithology controls how easily bedrock can be removed by geomorphic processes. Please clarify.

The Shield area is commonly described as being composed of resistant crystalline lithologies resulting in minimised glacial erosion and limited debris production (e.g. Shilts et al., 1987).

174. What is 'thin' and 'thick'? Please give some sort of quantification to adjectives used throughout the text.

We have clarified that thinner till veneer is typically < 2m and thicker till blankets are typically > 2m.

177. I believe the text in the caption is a bit too small. All the text in each map should be clearly readable, including the disclosures of where the data comes from.

We have made the text larger in all figures.

180. Is the study area with in the black polygon in (b)? if so this should be clearly stated.

It is stated in the figure caption and within the figure (identified as 'study area' in upper right of polygon) – this has been made larger so as not to miss it.

195. I believe these dashes should be longer?

We have made the dashes longer.

211. The sentence should not start with an acronym

Thank you, we have changed this to start as follows: 'High-resolution digital elevation data, made available through the ArcticDEM (10 m)…'

233. Adding a figure (or referring to, but I couldn't find a fig showing that, maybe Tab 2) that show the concept of the methodology explained here would be extremely useful. I think I understand where the authors are going, but with the current material I can't be sure.

We have now included an additional figure which shows the ice margin isochrones (Dyke et al., 2003) across the area and demonstrates how they are used as transects for sampling the meltwater routes.

282. Is it really a point if a width can be extracted? replace by 'location'?

We have changed this to location.

287. Please explain what the roughness at that scale is expected to be relevant for.

We have now updated the methodology and the resultant figure and have included a paragraph which justifies the parameter selection. This is based on the literature which discusses the transfer of bed topography to the surface (e.g. Gudmundsson, 2003; Ignéczi et al., 2018).

303. 'wider' than what? Or is it just to say vast majority eskers are part of MWroutes network?

We have rephrased this to indicate part of the same network.

314. The maps in Figure 3 have too large of a scale to actually make that point. Consider choosing smaller areas to make that point.

Here we meant that the updated meltwater routes map is denser than the initial esker map, suggesting that the holistic mapping captures a more complete picture of the drainage. We believe that this figure demonstrates this at the scale it is at.

315. updated compared to what? Just say clearly which map is referred to.

We have added in a sentence to explain that we have updated Storrar et al., (2013) mapping: 'Esker mapping by Storrar et al., (2013) was updated in the study area. Due to the higher resolution data available and the smaller spatial area covered, smaller features which may have been missed could be included.'

316. Wider than what?

We meant this in reference to the fact that features in the scale of 100s – 1000s m (i.e. meltwater channels, meltwater corridor or esker splays) are recorded at a vast majority of sample locations. We have made this clearer and removed the subjective description.

326. Define L and U quartiles.

We have done this.

326. Add centerline spacing, to make it more obvious to the reader.

We have done this.

329. Avoid starting sentence with acronym. Also the subject of the sentence should be the attribute of the feature (length or width) not the generic name of the feature itself.

We have done this and removed a significant number of the acronyms from the paper all together.

333. This isn't clear. Around what part of the ice sheet.

We have changed this to across the study area.

335. This needs to be clarified as well. The sentence talks to spatial changes, and the link between spatial and temporal involves some assumptions about landform formation.

We have added in the sentence: 'If these landforms are assumed to have formed time-transgressively, this would suggest no clear trend in width during deglaciation.'

337. What is that range? Note that the range from models is directly tied to the range

of conductivity values used in these models which in essence is rather arbitrary and tuned to match field observations. Perhaps, best to remove the models from this paragraph.

To make this clearer we have added in a new figure (modified from Storrar et al., 2014), which demonstrates the range of spacings presented in earlier studies and this one. Only one of the 11 samples included is from a model (the rest are observations), and this is included just for comparison.

339. Again, the step from spatial to temporal involves a number of assumptions and interpretation. It is best to keep to the spatial interpretation in the results and develop the importance for expected temporal evolution for the discussion.

We have kept this in the results section but have added 'if we assume a time transgressive formation' as we believe this is an important observation.

345. Please put number on each panel or make it clear that the figure is separated in 3 rows. The meaning of column is clear looking at the figure, not that of the rows.

We have added this in and now what each panel shows is clearer.

350. I can't find these locations in Fig. 2. Also the caption of Fig. 2 only points to Fig 4 and 5, not 3. Are these the unlabelled black squares in Fig.2? Please update with clear labels for each locations.

We have updated the figure to make it clearer where each location is.

359. I think section 4.2 can be removed. It repeats what has been said before and brings no new information.

We think that this is useful for contextualising this work so have included part of the paragraph but have consolidated it and combined it with the esker mapping section.

367. what does definition mean here? It's the first time it's used in this context. Please keep same names and descriptions as earlier or define terms.

We have changed this to make the description on relief quantifiable.

381. Please make sure the features you describe in the text to make your points are

visible in the figure. Esker and esker splays are really hard to visualize here, and in some cases hummocks too. Consider zooming in the panels or adding a figure if the whole area needs to be displayed for other purposes as well.

Thank you for pointing this out, we have redone this figure zooming in to key geomorphic signatures which are important in this paper. We have also put this figure in the introduction (now figure 2) so as to provide the reader with a clear understanding of what is being discussed from the start.

381. Use numbers in the topo elevation scale. High and low mean nothing here as the relief described here are all low compared to alpine or fjord type glacial landscapes.

The elevation scale bar has been removed as we now use greyscale hillshades to better visualise the features.

382. Add delineation of MWroutes on figure. Where is north? What were the local ice flow direction? Where are these regions from with respect to the general map? Are they the colored dots?

Please see answer to 381.

393. Many redundancies in this paragraph, please consolidate.

We agree and have consolidated this.

400. Where are these examples? Where is north? Give numbers for elevation scale It would be useful to the outlines of the landforms on the DEM as well to clearly see the topography and better understand the analysis. Ice flow vector and text should be larger.

We have now added lat/lon to each image, a north arrow and ice flow direction. The images have been redone using greyscale hillshade to better visualise the features.

420. Are these statistics for the whole study area? It would be great to have the location of the mean marked on the plot (in addition to the number given) and it might be useful to add the median as well given the nature of the populations.

Yes, they are from the whole study area, we have added this to the figure caption. We have also marked the mean and median on each plot.

432. Because of the redundancy of labels in the panels, it would make the figure if the same labels where only written once per column or row and that the font be made bigger. Is there a meaning for the different color used? if so please explain. The authors might also want to make a color choice that is suitable for people with colored impaired vision.

Thank you for this recommendation, we have updated the labels. We have used the colours blue to represent over representation and red to indicate under representation.

432. For consistency it would be useful to have the panels for each landform in the same order as in Fig. 6. Also each sub-panel should be labelled to make the back an forth between figure and text easier.

We agree and have changed the ordering of these panels to match the figure above.

442. I'm not sure I understand the display of the std dev of the topography. If a moving window is used, then that results in a map of topography std dev. So where dot the 1 to 5 std dev come from? Why is it not just the value of the std dev that is displayed on the map?

We have now changed this figure completely so there is just one map with the roughness standard deviation surface (please see comment 287 for parameter selection) with the meltwater routes overlain on this. This demonstrates that the area with the highest roughness (at a scale expected to be relevant for bed to surface transfer), coincides with the area with the densest subglacial meltwater routes.

446. I think it would be worth expanding a bit more than 2 sentences on figure 9.

We have done this.

450. what is the std window used in panel (a)? or is the coloring the density of MWroutes?

Please see comment 442.

450. This figure is too small. Where is north? What is the scale? perhaps put lat-long lines? Why not display the MWroutes on all panels?

Please see comment 442.

473. Figure needs to be bigger and font larger.

We have redone this figure. As well as making the figure and font larger, we have added an inset to show the distribution of ice streams across the Canadian Shield at a larger scale. We have also sampled meltwater route density in boxes across ice stream and non-ice stream areas and have compared these quantitatively using a two-sample t-test. This confirms that there is a significant difference between density of meltwater routes in and out of ice streams.

490. Make the colored line thicker, they are really hard to see.

We have made the coloured lines thicker.

490. Thank you for adding number on till cover! Currently the colors for veneer and blanket are hard to differentiate though.

We have changed the colours so they are easier to differentiate between.

491. Where do these example come from in other maps? Is the distance along the ice flow? why are not all the panels labelled? Why sample at 5km and not at shorter interval?

We have coloured the long profiles in our locations map (now Fig. 5) to show where the examples are from.

Yes, this is the distance along flow, we have added this to the figure axis to make this clearer.

We have now resampled along each meltwater route at 1 km intervals. This captures more of the variations in landform expression (i.e. meltwater track to meltwater corridor transitions) as well as the more fragmented esker ridges.

570. While the general idea is right, it necessary to break down and explain a bit more how these different papers and method infer or calculate large pressure in subglacial conduits. Also, the recent study by Rada and Schoof (2018, The Cryosphere) is perhaps the most comprehensive documentation of borehole water pressure to date and is worth discussing with regard to the context of the current study.

Thank you for this recommendation. We have now included this within the updated paper.

**References:**

Bartholomew, I. Nienow, P. Sole, A. Mair, D. Cowton, T. Palmer, S. Wadham, G. Supraglacial forcing of subglacial drainage in the ablation zone of the Greenland ice sheet. Geophysical Research Letters. 38(8). 2011.

[revised manuscript text omitted]